# Chromatinization modulates topoisomerase II processivity

Jaeyoon Lee[1], Meiling Wu [1,2], James T. Inman[1,2], Gundeep Singh[3], Seong ha Park[3], Joyce H. Lee [4], Robert M. Fulbright[1], Yifeng Hong[5], Joshua Jeong[4], James M. Berger [4] & Michelle D. Wang [1,2]✉

Type IIA topoisomerases are essential DNA processing enzymes that must robustly and reliably relax DNA torsional stress. While cellular processes constantly create varying torsional stress, how this variation impacts type IIA topoisomerase function remains obscure. Using multiple single-molecule approaches, we examined the torsional dependence of eukaryotic topoisomerase II (topo II) activity on naked DNA and chromatin. We observed that topo II is ~50-fold more processive on buckled DNA than previously estimated. We further discovered that topo II relaxes supercoiled DNA prior to plectoneme formation, but with processivity reduced by ~100-fold. This relaxation decreases with diminishing torsion, consistent with topo II capturing transient DNA loops. Topo II retains high processivity on buckled chromatin (~10,000 turns) and becomes highly processive even on chromatin under low torsional stress (~1000 turns), consistent with chromatin's predisposition to readily form DNA crossings. This work establishes that chromatin is a major stimulant of topo II function.

Topoisomerases are ubiquitous enzymes required for solving a variety of topological problems resulting from the double-helical structure of DNA[1–4]. Type IIA topoisomerases are of particular interest due to their ability to both relax supercoiled DNA and decatenate DNA molecules[5,6]. These enzymes act on DNA through a strand-passage mechanism, in which they hydrolyze ATP to pass one segment of DNA (transfer or T-segment) through a transient, enzyme-mediated double-stranded break in another segment (gate or G-segment)[5,6]. Notably, this action requires the enzyme to capture both G- and T-segments, necessitating that two DNA segments be in close proximity and form a crossing prior to strand passage.

Eukaryotic type IIA topoisomerases (topo II) are vital for the resolution of torsional stress during transcription and replication[2,3,7]. In vivo, torsional stress is dynamic, varying over both space and time. During transcription, RNA polymerase progression generates DNA supercoiling, which increases as transcription levels increase[8].

Torsion accumulated near active genes can impact cellular functions thousands of base pairs away by supercoil diffusion[9]. Similarly, during DNA replication, the degree of supercoiling near the replication fork varies over time, with torsional stress increasing towards termination[10]. Thus, for topo II to properly support supercoiling homeostasis and cell viability[11–14], it must relieve dynamically varying levels of torsional stress, yet whether or how the enzyme adjusts its activity in response to this variation is not fully understood. Previous biochemical and single-molecule studies have successfully elucidated many aspects of topo II activity on naked, plectonemically supercoiled ("buckled") DNA under high torsional stress[6,15–22]. However, torsion can also accumulate in "pre-buckled" DNA prior to buckling, and there is no clear assessment of topo II action on such a substrate. Even less is known about the impact of chromatinization on topo II activity, limiting our understanding of the enzyme in in vivo.

[1]Physics Department & LASSP, Cornell University, Ithaca, NY 14853, USA. [2]Howard Hughes Medical Institute, Cornell University, Ithaca, NY 14853, USA. [3]Biophysics Program, Cornell University, Ithaca, NY 14853, USA. [4]Department of Biophysics and Biophysical Chemistry, Johns Hopkins University School of Medicine, Baltimore, MD 21205, USA. [5]Department of Electrical and Computer Engineering, Cornell University, Ithaca, NY 14853, USA. ✉e-mail: mwang@physics.cornell.edu

Questions about the torsional response of topo II have been challenging to address due to experimental constraints. Previous techniques have been limited to studies of topoisomerase-mediated relaxation of DNA that are undergoing time-varying changes in topological state. In particular, single-molecule studies of topo II have only measured the relaxation of naked buckled DNA[15,18,19,21,22]. In addition, while topo IIs from various organisms have been shown to preferentially bind supercoiled DNA over relaxed DNA[23–25], how the processivity and strand passage rate of topo II depends on torsional stress remains unknown. Chromatin presents a significant set of challenges for studying topo activity, and while previous studies have indirectly probed topo II activity on chromatinized substrates[21,26,27], direct measurements of topo II's biophysical properties on chromatin have not been possible.

Here, we examine topo II relaxation of torsionally strained DNA in both buckled (plectonemically supercoiled) and pre-buckled (non-plectonemically supercoiled) regimes by developing single-molecule assays that enable direct measurement of topo II activity. We then apply these assays to chromatin to characterize the impact of chromatinization on topo II activity. Taken together, our results show that chromatinization modulates topo II activity by enhancing processivity,

enabling the enzyme to efficiently resolve torsional stress even prior to DNA buckling.

## Results

### Quantitative definition of torsional stress

To understand how topo II responds to different degrees of torsional stress, it is important to first quantitatively define this metric. Torque in DNA resists rotation/twist, so torsion can be characterized by torque (analogous to linear stress being characterized by force, which resists translocation)[28]. Using an angular optical trap (AOT) (Fig. 1a), we directly measured the torque necessary to add turns to DNA as a function of the number of turns added (DNA supercoiling state) using methods we established previously[21,29–32]. As shown in Fig. 1b, when turns are added to DNA held under a given force, torque (torsion) in DNA increases until the DNA buckles to extrude a plectoneme[31–33]. Before the buckling transition, a significant fraction of added turns is converted to DNA twist, while after the buckling transition, torque no longer increases with the further addition of turns, and all additional turns are converted to DNA writhe (Fig. 1b, c). Thus, torsional stress increases with added turns prior to buckling but remains constant after buckling.

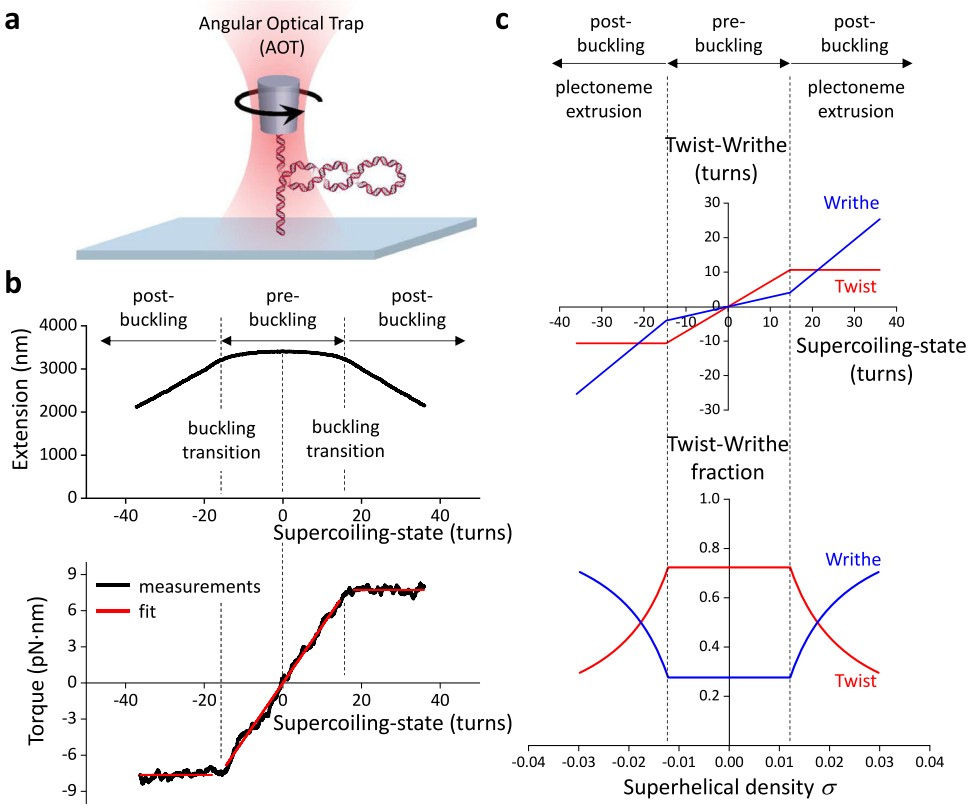

**Fig. 1 | Torsional response of naked DNA measured by the angular optical trap (AOT). a** Experimental configuration for measuring DNA torsional response on the AOT. A 12.7 kb DNA molecule containing a random sequence was torsionally constrained between a birefringent quartz cylinder and the coverslip surface of a microscope sample chamber. The AOT simultaneously measured and controlled the DNA extension, force, rotation, and torque. **b** Extension and torque versus turns added to DNA under constant force. Data were averaged from $N = 22$ biologically independent traces held under 0.5 pN force. In the pre-buckled regime, torque increased linearly as turns were added. The linear fit within the pre-buckled regime (middle red line) yields an effective twist persistence length of 78.9 ± 0.3 nm, consistent with previous measurements within measurement uncertainty[32]. At the buckling transitions (vertical dashed black lines), DNA underwent phase transitions to form plectonemes. The torque plateaued with further plectoneme extrusion at +7.7 ± 0.2 pN•nm (mean ± s.d) for the (+) supercoiled DNA and −7.6 ± 0.3 pN•nm for

the (−) supercoiled DNA (horizontal red lines). **c** Partitioning of twist and writhe. The DNA supercoiling state is defined by the turns added to DNA, or the change in the linking number $\triangle Lk = Lk - Lk_0$, with $Lk_0$ being the linking number of torsionally relaxed DNA. $\triangle Lk$ is partitioned into a change in twist $\triangle Tw$ (torsion) and a change in writhe $\triangle Wr$: $\triangle Lk = \triangle Tw + \triangle Wr$. The superhelical density $\sigma$ is used to characterize the degree of supercoiling: $\sigma = \frac{Lk - Lk_0}{Lk_0} = \frac{\triangle Lk}{Lk_0}$. Using the fit parameters from (**b**), the partitioning of added turns into the twist and writhe was obtained. In the pre-buckling regime, the added turns partition mostly into twist, which increases linearly with turns; the partitioning of twist was calculated from the ratio of the effective twist persistence length, 78.9 nm, and the intrinsic twist persistence length, 109 nm[32]. Beyond the buckling transition, all added turns partition into writhe, thus maintaining a constant torque. Source data are provided as a Source Data file.

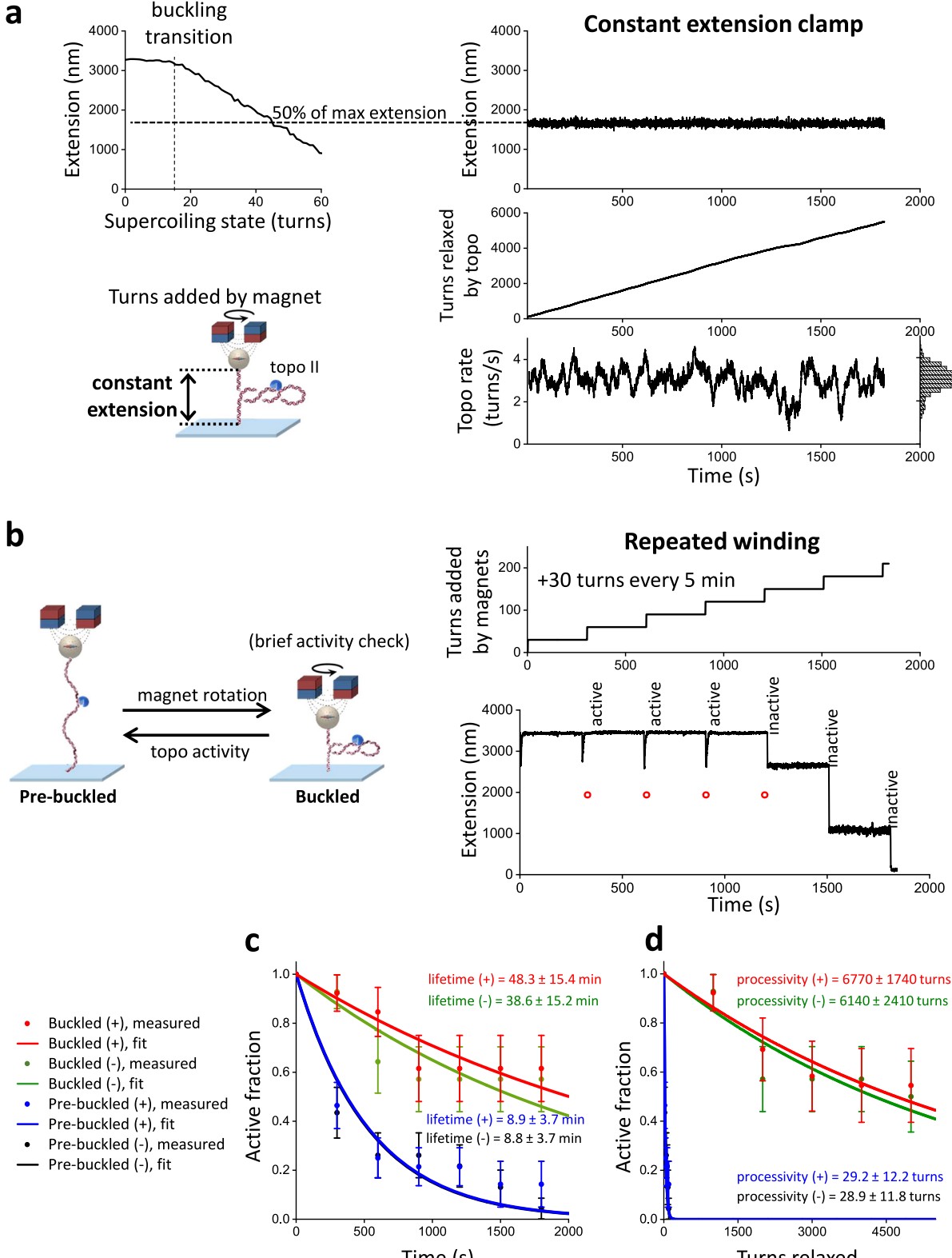

## Topo II is highly processive on buckled DNA

We first investigated topo II processivity on buckled and pre-buckled DNA (Fig. 2). To enable direct measurement of topo II processivity and relaxation rate on a buckled DNA in a constant topological state, we developed a single-molecule constant-extension method to clamp the DNA extension (maintain a constant superhelical density) as topo II continuously relaxed the substrate. We implemented this method on a

custom-built magnetic tweezers (MT) instrument (Fig. 2a; Supplementary Fig. 1; Methods). To measure the activity of a single topo II enzyme on a DNA molecule, we pre-incubated the chamber with a low concentration of yeast topo II and flushed excess topo from the chamber (Methods). After topo II binding, we used the extension clamp and kept the DNA in a post-buckled state via magnet rotation, whose rate was equal and opposite to the topo II relaxation rate and

**Fig. 2 | Plectonemes stabilize topo II activity on DNA. a** Measurements of topo II activity lifetime, processivity, and relaxation rate on buckled DNA in a constant topological state. 12.7 kb DNA molecules containing a random sequence were torsionally constrained between magnetic beads and the coverslip surface of a microscope sample chamber, and initial extension-supercoiling state relations were measured under 0.5 pN force (top left). A DNA molecule showing topo II activity was selected, wound to 50% of its zero-turn extension, then held at this extension for 30 min via magnet rotation counteracting topo activity (right). The topo relaxation rate was calculated from the magnet turns filtered by a 20 s window; the rate distribution is shown as a histogram (bottom right). **b** Measurements of topo II activity lifetime and processivity on pre-buckled DNA. +30 turns were added every 5 min to DNA molecules under 0.5 pN force by rapid magnet rotation (40 turns/s) (top right). After each winding step, DNA molecules with active topo II were identified by increases in extension; molecules without topo II activity

remained at a constant extension. A representative trace with topo II activity for the first four winding steps is shown (bottom right). Red circles indicate expected extension if topo II was not able to relax pre-buckled DNA; if the extension immediately after a winding step was above a red circle, then topo II had relaxed DNA into the pre-buckled regime during the previous relaxation step. **c** Activity lifetime of topo II on (+) buckled (red), (-) buckled (green), (+) pre-buckled (blue), and (-) pre-buckled DNA (black) with $N = 13, 14, 28$, and 23 biologically independent traces, respectively; error bars represent standard errors of the proportions. These data were fit to exponential curves to estimate the lifetimes. **d** Processivity of topo II on (+) and (-) buckled DNA and (+) and (-) pre-buckled DNA from the same set of traces considered in Fig. 2c; error bars represent standard errors of the proportions. These data were fit to exponential curves to estimate the processivities. Source data are provided as a Source Data file.

thus served as an excellent readout of topo II activity. This constant-extension method effectively creates an infinitely renewable substrate maintained in a constant buckled state, permitting the measurement of topo processivity well beyond that attainable in a conventional supercoiling relaxation experiment while also guaranteeing that the DNA always remains well within the buckled regime.

Unexpectedly, we discovered that yeast topo II was extremely processive on buckled DNA. Figure 2a shows a constant-extension measurement with a single yeast topo II that remained active for 30 min, at which point the experiment was manually terminated. During this time, the topo II removed 5500 turns from the DNA. Strikingly, this processivity is ~50 times greater than those determined from previous measurements for human topo IIα[18]. To determine whether this difference might be due to the species-specific properties, we also applied the constant-extension method to human topo IIα and human IIβ (Fig. 3). As with yeast topo II, both human topo II enzymes were also highly processive, capable of relaxing thousands of turns without dissociation. Thus, high processivity on buckled DNA appears to be a relatively general feature of eukaryotic type IIA topoisomerases.

In addition to processivity, each trace provided a measurement of the instantaneous relaxation rate from a single topo II. The rate of relaxation in this trace was homogeneous at 3.0 turns/s on average, with some modest variation (coefficient of variation of 0.19) that could reflect the stochastic nature of topo II activity (Fig. 2a, bottom right). This variation is also consistent with enzyme-to-enzyme variations (Supplementary Fig. 1e). Because topo II changes the linking number (Lk) of DNA in steps of 2 with each reaction cycle, this value indicates that the enzyme is catalyzing 1–2 rounds of strand passage per second, consistent with previous biochemical measurements[34].

## Topo II is less processive on pre-buckled DNA

Although topo II activity is extremely processive on buckled DNA, it was unclear whether topo II is similarly active on pre-buckled DNA, which does not contain a plectoneme to provide suitable G- and T-segments in close proximity. Unfortunately, detecting the relaxation of pre-buckled DNA is challenging, as any strand passage event by topo II will lead to only a minimal DNA extension change. To circumvent this limitation, we performed a "repeated winding" experiment to measure topo II activity on pre-buckled DNA (Fig. 2b). For this approach, we repeatedly checked for topo II activity by periodically adding a small number of turns to the DNA to bring the molecules slightly into the buckled regime. If topo II was active, the extension of the tether increased as the topo II relaxed the newly added supercoils.

The repeated winding study revealed that topo II can relax pre-buckled DNA. Figure 2b shows a typical trace, in which +30 turns were added every 5 min by rapidly rotating the magnets at 40 turns/s. Topo II remained active for three additional rewinding steps after the initial winding step, as demonstrated by the increasing extension. The extension after each of those rewinding steps corresponded to

the expected extension if, at the start of the rewinding step, the DNA were near the fully relaxed state instead of at the buckling transition. This result indicates that in the time between successive rewinding steps, topo II relaxed all plectonemic supercoils and continued to relax the supercoiling in the pre-buckled DNA to a near-fully relaxed state. To our knowledge, processive relaxation of pre-buckled DNA by a type IIA topoisomerase to near completion has not been previously reported. The DNA was buckled for only a small duration (0.75 s) during the rewinding step in which topo II relaxed at most 2 turns, so relaxation of the added turns occurred primarily in the pre-buckled state. In addition, the 5 min of topo II relaxation in between rewinding steps was insufficient for complete relaxation (Supplementary Fig. 2), so topo II was continuously relaxing supercoiling as long as it remained active. Thus, the activity lifetime measured by this experiment accurately reflects that of topo II on pre-buckled DNA, i.e., for the trace shown in Fig. 2b, topo II retained activity on the pre-buckled DNA for 15–20 min.

We next compared statistics from multiple traces obtained with (+) supercoiled buckled and pre-buckled DNA to examine the effect of torsional stress on topo II processivity (Fig. 2c, d). On buckled DNA, we did not observe any permanent loss of activity before 30 min, although 40% of traces contained long pauses (>30 s duration) that may be due to bound topo II becoming temporarily inactive before becoming active again (Fig. 3). It is also possible that the bound topo II became permanently inactive or dissociated and relaxation resumed after the binding of another topo II, but we expect such events to be extremely rare (Methods). Nonetheless, when calculating the fraction of traces with continued topo activity ("active fraction") as a function of time, we conservatively classified individual traces as being "active" only until the first pause to avoid overestimating the processivity. Fitting the active fraction over time to an exponential yielded a lower bound of $48.3 \pm 15.4$ min for the mean activity lifetime. By considering the active fraction versus turns relaxed, we also obtained a lower bound of $6770 \pm 1740$ turns for the processivity. This lower bound (>6000 turns) far exceeds values suggested by other single-molecule methods for similar enzymes[15,18,19]. Our results with human topo IIα and IIβ (Fig. 3) suggest that this discrepancy could arise from differences in methodology, although differences in enzyme preparations and buffer conditions may also contribute.

In contrast with the relaxation of buckled DNA, our repeated winding experiment showed that topo II associates less stably with pre-buckled DNA, with a mean activity lifetime of $8.9 \pm 3.7$ min. This result may be a slight overestimate because the bound topo II could have dissociated and been replaced by another topo II in between successive rewinding steps, which would be indistinguishable from the same enzyme remaining bound in our experiment. However, we again expect such events to be rare because free topo II was flushed from the sample chamber (Methods), and this possibly overestimated value is still far smaller than the activity lifetime of topo II on buckled DNA. Since the DNA extension immediately after each rewinding step

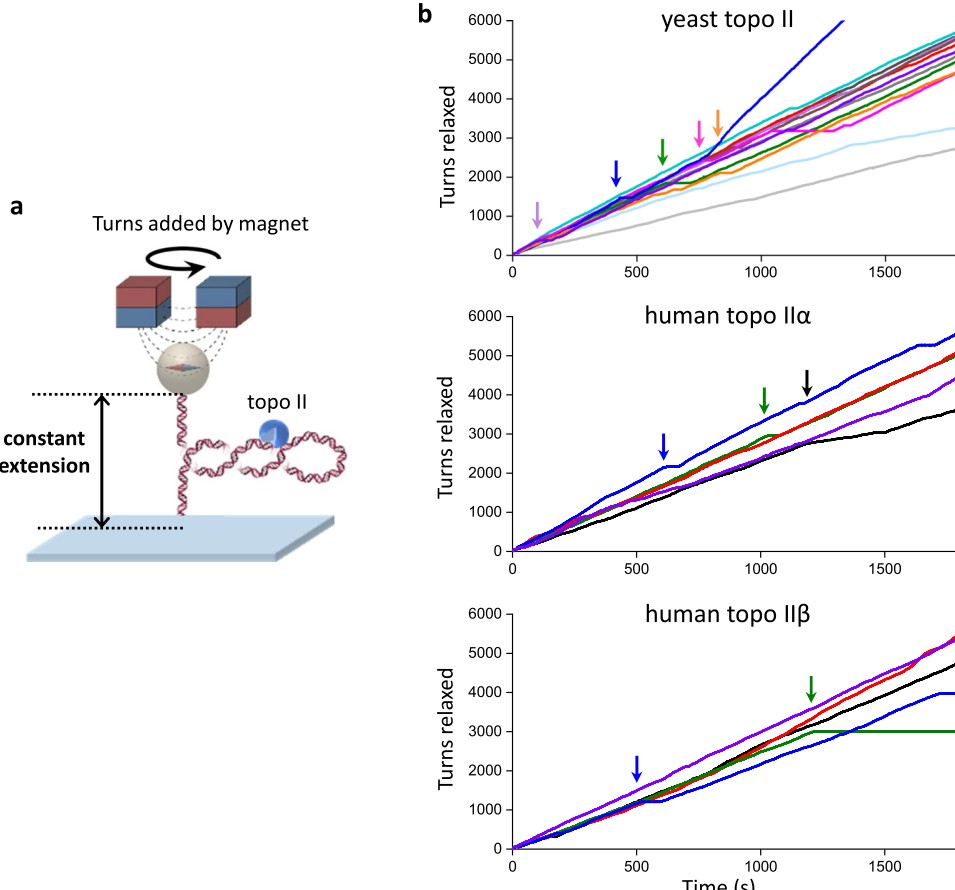

**Fig. 3 | Human topo IIα and IIβ are also highly processive on buckled DNA.**
**a** The constant-extension experimental configuration, as shown in Fig. 2a (reshown here for clarity). All experiments shown in this figure were conducted on the 12.7 kb DNA clamped at 50% extension under (+) supercoiling. **b** Topo II activity traces of turns relaxed vs. time: yeast topo II (top panel, same traces as those used in Fig. 2c, d), human topo IIα (middle panel), and human topo IIβ (bottom panel) with $N = 13$, 5, and 5 biologically independent traces, respectively. Each color represents a measurement from a single DNA molecule, with an arrow of the same color indicating the first occurrence when the topo II paused for >30 s. For the dark blue trace of yeast topo II, topo II relaxation paused with duration >30 s at 430 s (and thus classified as no longer active at that time) and later showed a doubling of the rate beginning at 770 s, likely due to tandem relaxation by two topo II molecules. Source data are provided as a Source Data file.

indicated the DNA supercoiling state in the pre-buckled regime before rewinding (Supplementary Fig. 2), we could also estimate the processivity of topo II in the pre-buckled regime by plotting the active fraction as a function of the cumulative number of turns applied to the DNA in the pre-buckled regime. Fitting to an exponential yielded a mean processivity of $29.2 \pm 12.2$ turns, a number far smaller than the lower bound for topo II processivity on buckled DNA ($6770 \pm 1740$ turns). To assess whether the activity lifetime or processivity of topo II show any dependency on the chirality of the torsional stress (i.e., over- vs. under-winding), we repeated the same activity lifetime and processivity measurements by applying (-) supercoiling to the DNA. Our results indicate that there is no significant dependence of topo II activity lifetimes and processivities on the chirality of supercoiling applied to naked DNA (Fig. 2c, d), in good agreement with a previous biochemical study[17].

These findings establish that topo II is much more stably associated with buckled DNA compared to pre-buckled DNA. Topo II can relax many thousands of turns in buckled DNA before losing activity, ~50 times greater than previous estimates. In contrast, on pre-buckled DNA, processivity is reduced by at least ~100 fold. These findings may provide an explanation for previous studies which showed that type IIA topoisomerases preferentially interact with supercoiled DNA compared to relaxed DNA[23–25] and indicate that the increased affinity could stem from a marked reduction in the dissociation rate.

## Topo II rate slows down on pre-buckled DNA

Torsional stress in a DNA substrate could also influence topo II's relaxation rate. We found that the relaxation rate of topo II was constant on buckled DNA, independent of the plectonemic length (i.e., superhelical density) (Supplementary Fig. 3). Conversely, while Fig. 2b shows that topo II can relax pre-buckled DNA, it was unclear how the relaxation rate might depend on the DNA superhelical density and torsion in this regime. We therefore developed a method to investigate topo II's ability to relax DNA in the pre-buckled regime (Fig. 4a). We first buckled a DNA molecule with a single bound topo II by adding a fixed number of turns then monitored the DNA extension as it increased over time due to supercoiling relaxation. Once the DNA extension reached the buckling transition, we allowed for a specified duration of further topo II relaxation into the pre-buckled regime. We then performed a "buckling check" by applying an additional +35 rewinding turns to bring the DNA back into the buckled state, whose extension provided the final supercoiling state of the preceding topo II relaxation step (Methods). We repeated this process with different durations of topo II relaxation, from 2 s to 15 s, and tracked the relaxation of pre-buckled DNA over time by measuring the extension after each buckling check. For each trace, this was repeated either until topo II lost activity or to a maximum of 5 cycles. The rewinding was brief (0.875 s) with topo II relaxing only ~2.5 turns on average during each rewinding step, which we accounted for (Methods).

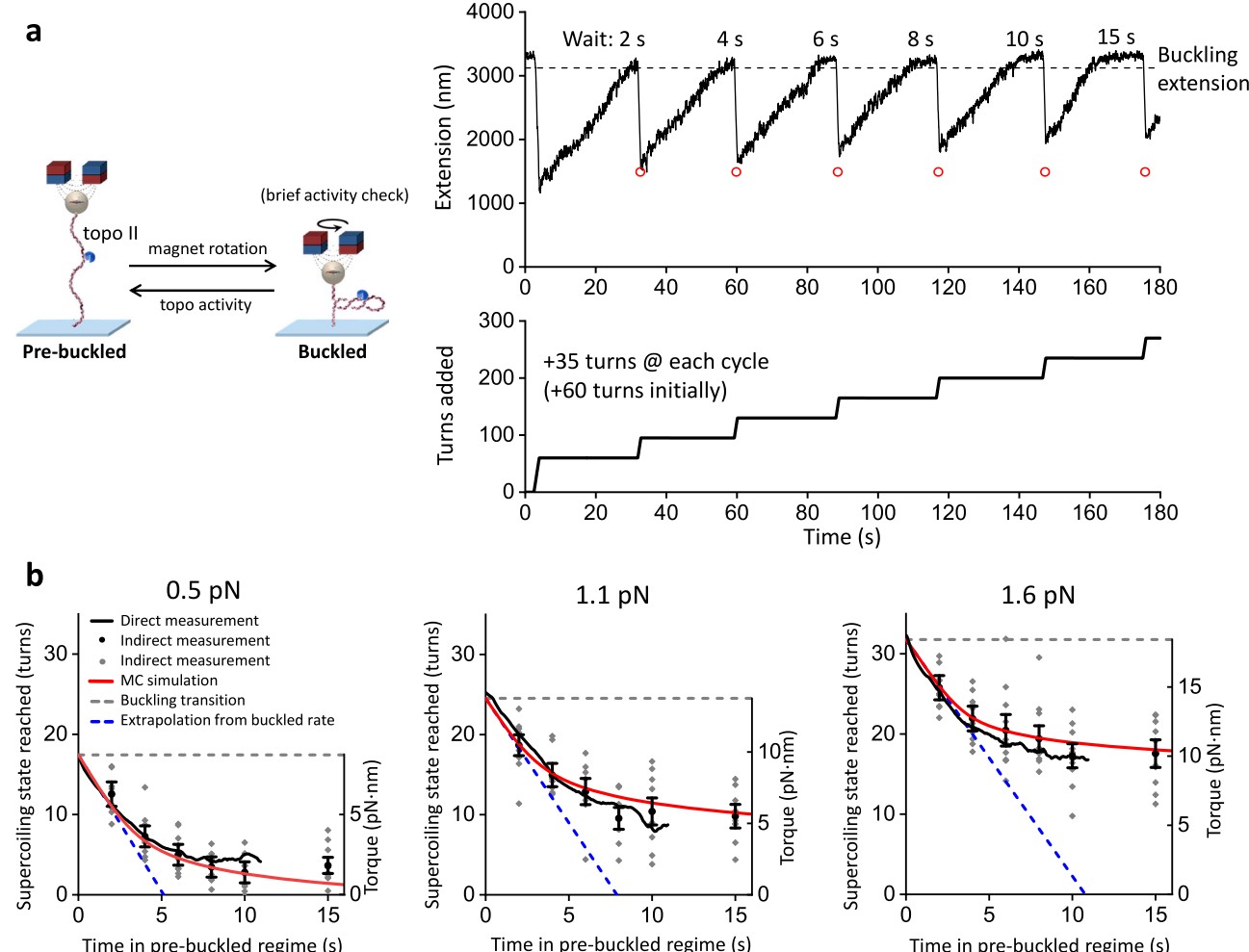

**Fig. 4 | Topo II captures spontaneous loops in pre-buckled DNA.**
**a** Measurements of topo II relaxation of pre-buckled DNA. Initially, +60 turns were added by rapid rotation of the magnets (40 turns/s) to a 12.7 kb DNA molecule containing a random sequence and with an active topo II. Once the extension reached the buckling transition due to topo II activity, a set duration of activity in the pre-buckled regime was allowed before +35 additional turns were added. If the extension immediately after a winding step was above the red circle (expected extension after winding if relaxed only to the buckling transition), then topo II had relaxed DNA into the pre-buckled regime during the previous relaxation step. The process was repeated for different durations. **b** Supercoiling state reached by topo II relaxation into the pre-buckled regime versus time of topo II relaxing the pre-buckled regime under 0.5 pN force, 1.1 pN, and 1.6 pN with $N$ = 10, 9, and 11 biologically independent traces, respectively. We calculated the DNA supercoiling state using two methods. In the direct-measurement method, the relaxation time-course during the 15 s wait shown in (**a**) was used to directly determine supercoiling state versus wait time in the pre-buckled state (Methods). In the indirect-measurement method, the supercoiling state was determined based on the shift in the extension-supercoiling state relation before and after topo relaxation in the pre-buckled state (Methods). For the indirect method, we show a scatter plot of all data points (gray) along with their means and SEMs for each duration of topo II activity on pre-buckled DNA (black). The torque in DNA corresponding to a supercoiling state is shown as the right vertical axis based on previous torque measurements[32]. The dashed blue lines are linear extrapolations using the relaxation rates on buckled DNA (Methods) and project the supercoiling state progression if enzyme activity remained constant into the pre-buckled regime. The red curves are theoretical results from a simple model assuming that topo II relaxation rate depends on the DNA loop formation rate obtained from Monte Carlo simulations (Methods). Source data are provided as a Source Data file.

As shown in Fig. 4b, the relaxation of pre-buckled DNA by topo II was initially as efficient as the relaxation of buckled DNA. However, topo II activity slowed down sharply as the DNA torsion decreased. To check whether topo II activity stopped completely at the observed plateau, we performed a separate experiment that allowed topo II to relax pre-buckled DNA for 10 min (Supplementary Fig. 2) and found that topo II further relaxed the pre-buckled DNA but not to completion, suggesting a slow rate towards full relaxation.

This method provides an indirect measurement of the progression of topo II relaxation in the pre-buckled regime. We also directly converted the extension data to the supercoiling state of DNA versus time in the pre-buckled state using the DNA extension-supercoiling state relation (Methods). As shown in Fig. 4b, the supercoiling state determined using the direct method agrees with that from the indirect method within the uncertainty of the measurement. This agreement

provides strong evidence that topo II can processively relax DNA into the pre-buckled regime to near completion, with the relaxation rate slowing down as the DNA torsion decreases.

## Topo II captures spontaneous loops in pre-buckled DNA

We hypothesized that topo II relaxes pre-buckled DNA in the absence of a plectoneme by capturing spontaneous loops that form as the DNA configuration thermally fluctuates due to DNA bending and writhing[32,35,36]. The incidence of such loops should depend on the DNA supercoiling state, with loops forming more often near or after the buckling transition than on more relaxed DNA[35]. As topo II relaxes DNA toward the fully relaxed state, spontaneous loops may become limiting, leading to the observed reduction in relaxation rate.

If this hypothesis is correct, then topo II should be less effective relaxing pre-buckled DNA that is held under a greater force[32,35,37].

Therefore, we examined topo II activity on pre-buckled DNA under different forces (0.5 pN, 1.1 pN, and 1.6 pN) (Fig. 4b). We observed that the number of turns left in the DNA after 10–15 s of topo II activity in the pre-buckled regime increased with increasing force, consistent with a reduction in spontaneous loop formation at a higher force. Interestingly, at a given supercoiling state, topo II relaxation rate slightly decreased instead of increasing with the DNA torsion (torque) when the force was increased, suggesting that the pre-buckled relaxation rate is more sensitive to DNA supercoiling state than changes in torsion.

Furthermore, spontaneous loop formation should decrease for a shorter DNA molecule[35]. Thus, we repeated the experiment with shorter DNA molecules and found that the relaxation of pre-buckled DNA is relatively slower (Supplementary Fig. 4). We also considered a simple writhe model previously used for topo II relaxation of post-buckled DNA by assuming that the topo II relaxation rate is only dependent on writhe according to the Michaelis-Menten relation[19]. We found that such a model cannot explain our measurements (Supplementary Fig. 5), suggesting that DNA writhing alone cannot explain topo II activity in the pre-buckled regime. Thus, while DNA writhe and the number of DNA loops are related, these two quantities are fundamentally not the same.

To relate topo II activity in a pre-buckled regime to DNA loop formation, we performed Monte Carlo simulations to compute the rate of DNA crossing formation as a function of DNA supercoiling state and force (Supplementary Fig. 6; Methods). For each supercoiling state under a given force, we generated equilibrium configurations of pre-buckled DNA using the parameters shown in Supplementary Table 1. From these configurations, we calculated the rate of DNA crossing formation and modeled the topo II relaxation rate by assuming it depends on the DNA crossing rate via a Michaelis-Menten type relationship (Methods). The results of these simulations show excellent agreement with our measurements (Fig. 4b). Because yeast topo II is known to introduce a bend in DNA[38,39], we also repeated the simulations by including a bend to examine how DNA bending may affect the crossing rate. These simulations also show good agreement with our measurements (Supplementary Fig. 7) when the increased formation of DNA loops is considered (Methods). Taken together, our data indicate that topo II relaxes pre-buckled DNA by capturing spontaneous DNA loops.

**Chromatinization enhances topo II processivity**

Chromatin is the natural substrate of eukaryotic topo II in vivo. Although there is some evidence that the C-terminus of topo II can associate with histone proteins[40,41], the catalytic domains of topo II likely interact directly with linker DNA between nucleosomes. Since chromatinization markedly alters the structural and torsional mechanical properties of DNA[21,42], we speculated that the activity of topo II on chromatinized substrates may be significantly different from that on naked DNA. In particular, chromatin exhibits an asymmetric extension-supercoiling state relation indicative of three distinct topological states[21]: two steep regimes on the left and right sides (the (-) and (+) buckled regimes, respectively), and a middle plateau regime with a slight negative slope (Fig. 5a, top). The presence of the extension plateau indicates that pre-buckled chromatin can effectively buffer (+) torsional stress, absorbing significant (+) DNA supercoiling without torsional build-up[21,43,44]. Consistent with this, the measured torque under (+) supercoiling shows a greatly reduced torsional stiffness in the extension plateau regime (Fig. 5a, bottom).

To examine the effect of chromatinization on topo II activity on buckled substrates, we used the constant-extension method (Figs. 2a and 3) with chromatin fibers containing ~50 nucleosomes assembled on a DNA construct comprising 64 repeats of a nucleosome positioning element (Fig. 5b; Methods). For this measurement, we pre-bound single topo II enzymes to chromatin fibers that satisfied our

selection criteria (Supplementary Fig. 8; Methods) and used the extension clamp to keep the fibers in the (+) or (-) buckled regime for up to 30 min (Methods).

As with buckled naked DNA, we found that topo II was extremely processive on buckled chromatin substrates. Figure 5b shows all traces of the constant-extension measurements on chromatin fibers, in both the (+) and (-) buckled states. As before, most traces did not show any permanent loss of activity, although a few contained long pauses (>30 s duration). We conservatively classified individual traces as being "active" until only the first pause, and fitting the active fractions over time to exponentials yielded lower bounds on the mean activity lifetimes of $112 \pm 29$ min in the (+) buckled state and $169 \pm 96$ min in the (-) buckled state (Fig. 5c). Similarly, we obtained lower bounds on the processivities of $12400 \pm 4700$ turns in the (+) buckled state and $22400 \pm 7600$ turns in the (-) buckled state (Fig. 5d). Comparison with the processivity for buckled naked DNA (Fig. 2d) suggests that chromatinization enhances topo II processivity on buckled substrates. The activity lifetimes and processivities are similar for (-) and (+) buckled states of chromatin, suggesting that the chirality of applied supercoiling is insignificant.

We then examined the effect of chromatinization on topo II activity under low torsional stress using the repeated winding experiment (Fig. 2b) with chromatin fibers under 0.5 pN force (Fig. 6; Methods). For this measurement, we pre-bound topo II to chromatin fibers that satisfied our selection criteria (Supplementary Fig. 8; Methods) and repeatedly checked for topo II activity by adding or removing a small number of turns (Methods). Figure 6a shows a typical trace of the repeated winding experiment with chromatin in the (+) winding direction, in which +20 turns were added every 5 min and topo II remained active for the full 30 min of the measurement. Chromatin fibers were mostly stable throughout this experiment, with only 1 nucleosome lost on average in 30 min (Supplementary Fig. 9).

As shown in Fig. 6b, c, even under low torsional stress, chromatin supported long activity lifetimes ($267 \pm 164$ min for (+) winding; $214 \pm 100$ min for (-) winding) and processivities ($1070 \pm 660$ turns for (+) winding; $854 \pm 402$ turns for (-) winding), both of which are ~20-fold greater than their naked DNA counterparts (compare Fig. 6 and Fig. 2). These results demonstrate that chromatin stabilizes topo II activity under low levels of torsional stress. Unlike a torsionally relaxed naked DNA with fewer DNA crossings, a torsionally relaxed chromatin fiber contains DNA crossings because of the juxtaposition of the entry and exit DNA segments at each nucleosome[45], which may stabilize topo II activity on pre-buckled chromatin.

**Topo II relaxation of chromatin is influenced by compaction**

The existence of three distinct topological states in the chromatin extension-supercoiling state relationship suggests that the relaxation rate of topo II may depend on the topological state of a chromatin substrate; however, this dependence has not yet been measured directly. Single-molecule methods used in previous studies are not suitable for such measurements because of the nonlinearity in the buckled regimes combined with the shallowness of the slope in the extension plateau regime[21]. To directly measure topo II relaxation rates in these three regimes of chromatin, we used the constant-extension method with chromatin fibers containing ~50 nucleosomes (Fig. 7a; Methods). For each trace, a chromatin fiber that satisfied our selection criteria (Supplementary Fig. 8; Methods) was bound to a single molecule of topo II and held for 2 min each in: (1) the (-) buckled regime at 50% of its maximum extension, (2) the midpoint of the extension plateau regime, and (3) the (+) buckled regime at 50% of its maximum extension. The mean rate of topo II relaxation in each state was calculated from the turns added during the corresponding clamping step.

We found that the magnitudes of the topo II relaxation rates in the (+) and (-) buckled regimes were not significantly different from each

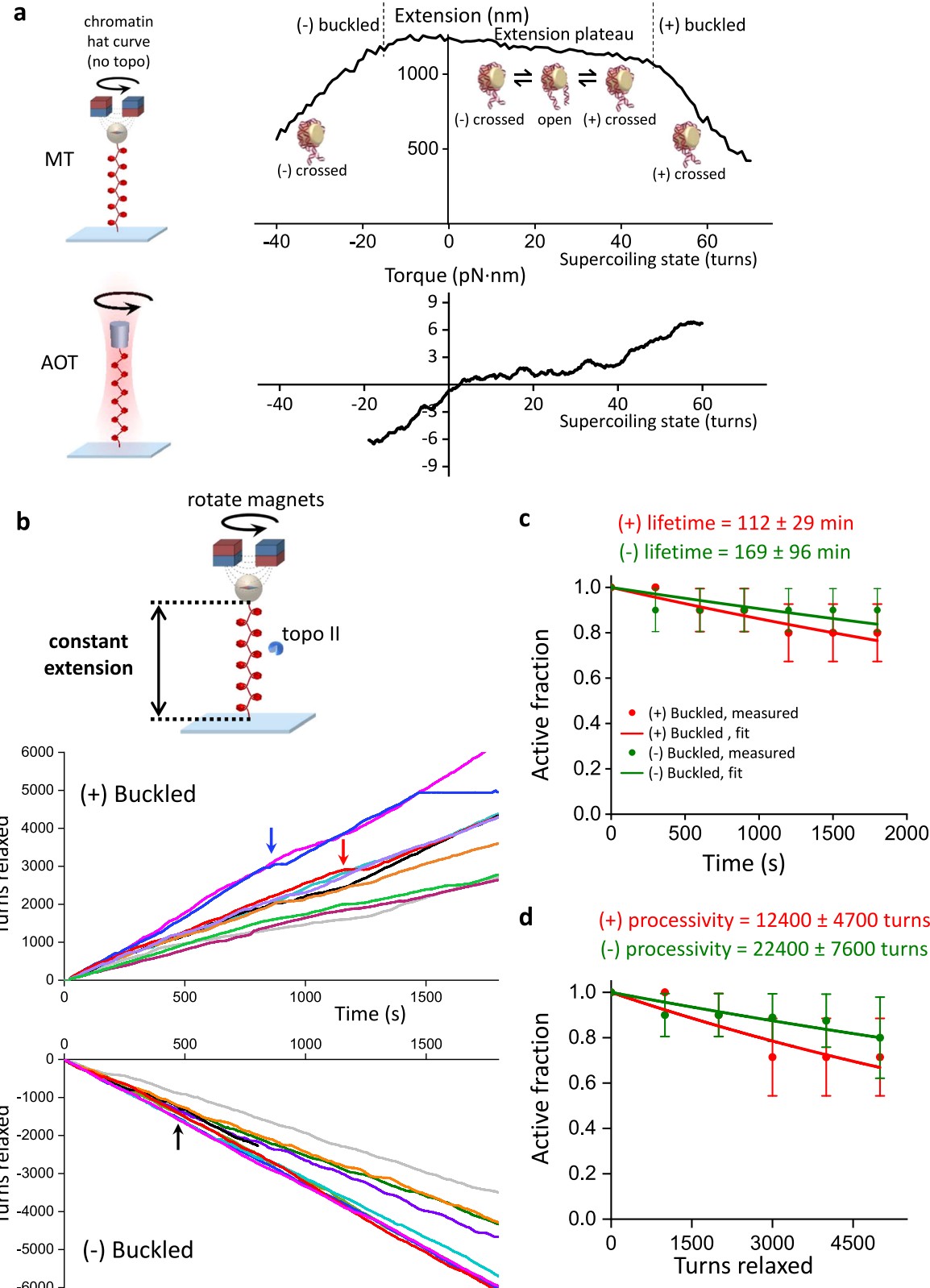

other, but the rate was slower in the extension plateau regime than in the buckled regimes by ~40% (Fig. 7b). This result suggests that DNA crossings were less available for topo II activity on chromatin in the less compact extension plateau regime. On buckled chromatin, the mean rate was consistent with that on buckled naked DNA (Supplementary Figs. 1 and 3), indicating that topo II relaxation was not limited by the availability of DNA crossings.

Interestingly, our results suggest that both the shape of the chromatin extension-supercoiling state relationship and the rate of topo II activity on chromatin are dictated by the left-handed solenoidal DNA wrapping within nucleosomes[45]. Due to this chirality, the entry and exit DNA segments of an individual nucleosome can cross in a supercoiling-dependent manner, with three distinct states: a (+) or (-) state, in which the entry and exit DNA segments cross with the

**Fig. 5 | Chromatinization preserves high topo II processivity on buckled substrates. a** Measurements of chromatin extension-supercoiling state relation before introducing topo II. Chromatin fibers were assembled on 12.7 kb DNA containing 64 repeats of a nucleosome-positioning element and were torsionally constrained between magnetic beads and the coverslip surface of a microscope sample chamber under 0.5 pN force. The top panel is a representative extension-supercoiling state relation for a chromatin fiber containing 48 nucleosomes. Shown in the bottom panel is the corresponding torque-supercoiling state relation previously measured using the angular optical trap[21]. **b** Topo II activity traces of turns relaxed vs. time. Data were acquired using the constant-extension method at 50% of the maximal extension of the chromatin fiber under 0.5 pN force. These experiments were performed the same way as shown in Figs. 2a and 3 but with chromatin

fibers instead of naked DNA. Each color represents a measurement from a single chromatin fiber, with an arrow of the same color indicating the first occurrence when the topo II paused for >30 s. For the black trace under (-) buckling, topo II relaxation paused at 260 s, and the trace was manually terminated at 800 s when another free magnetic bead in the sample drifted over the tether and obstructed the bead position detection. **c, d** Activity lifetime and processivity of topo II on buckled chromatin under high (+) torsional stress and high (-) torsional stress, from data shown in (**b**), with $N$ = 10 biologically independent traces for each condition; error bars represent standard errors of the proportions. These data were fit with exponential curves to estimate the activity lifetimes and processivities. Source data are provided as a Source Data file. Images in 5a and 5b are reprinted from ref. 21, with permission from Elsevier.

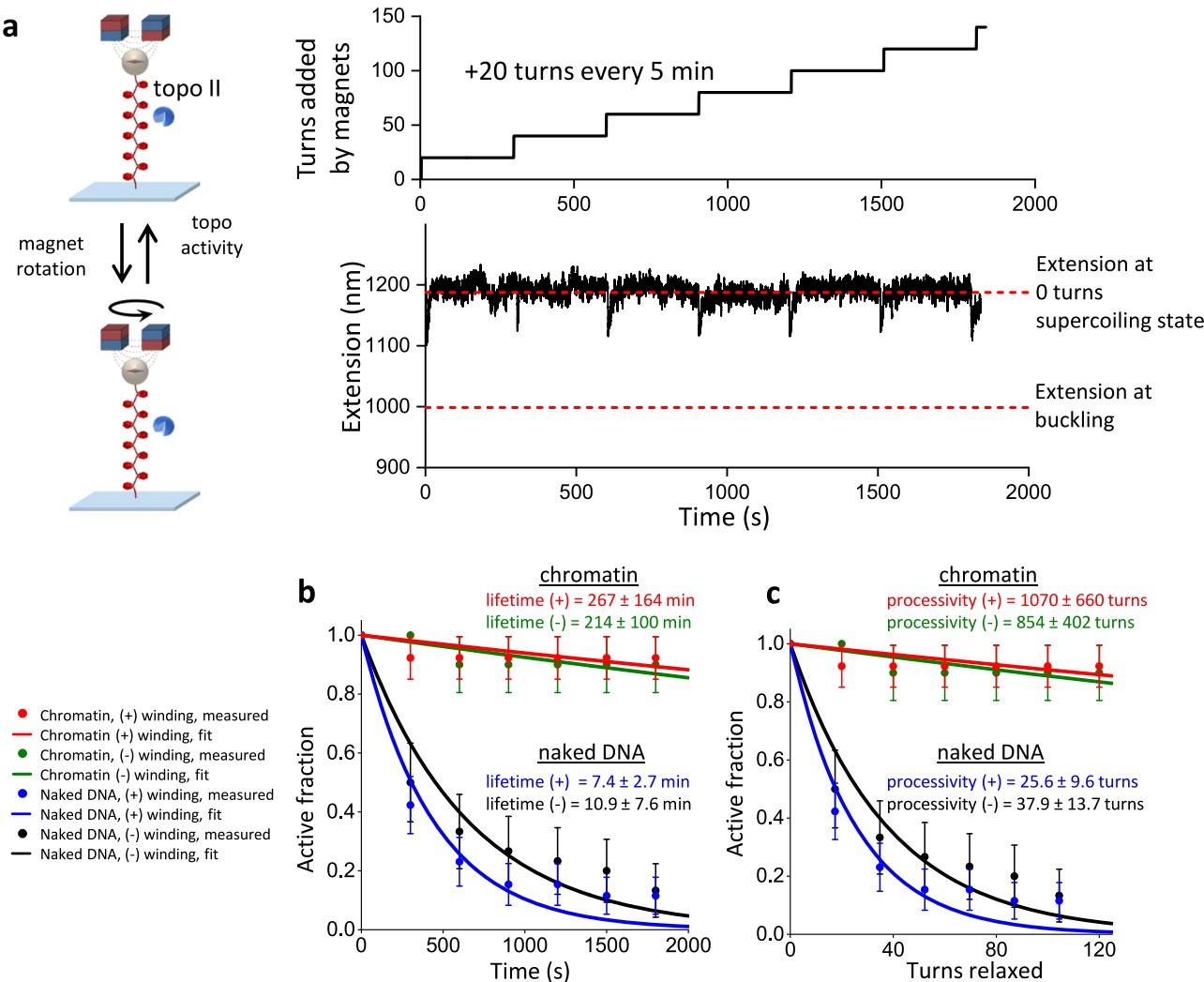

**Fig. 6 | Chromatinization enhances topo II activity under low torsional stress. a** Measurements of topo II activity lifetime and processivity on chromatin under low torsional stress. 20 turns were added every 5 min to chromatin fibers under 0.5 pN force by rotation of the magnets. After each winding step, chromatin fibers with active topo II were identified by increases in extension. In the example trace, the chromatin fiber was under (+) winding, and its extension was consistently greater than that at the buckling (the lower dashed red line) for the entire active time duration, indicating that topo II relaxation occurred within the pre-buckled regime. **b** Activity lifetime of topo II on chromatin under low (+) torsional stress and low (-) torsional stress under 0.5 pN force with $N$ = 13 and 10 biologically independent traces, respectively; error bars represent standard errors of the proportions.

These data were fit to exponential curves to estimate the activity lifetimes. For comparison, we also show the corresponding data and fits for the same DNA substrates containing 64 repeats of a nucleosome-positioning element but without nucleosomes under low (+) torsional stress and low (-) torsional stress with $N$ = 28 and 30 biologically independent traces, respectively. **c** Processivity of topo II on chromatin under low (+) and (-) torsional stress from the same set of traces considered in (**b**); error bars represent standard errors of the proportions. These data were fit to exponential curves to estimate the processivities. Source data are provided as a Source Data file. Image in 6a is reprinted from ref. 21, with permission from Elsevier.

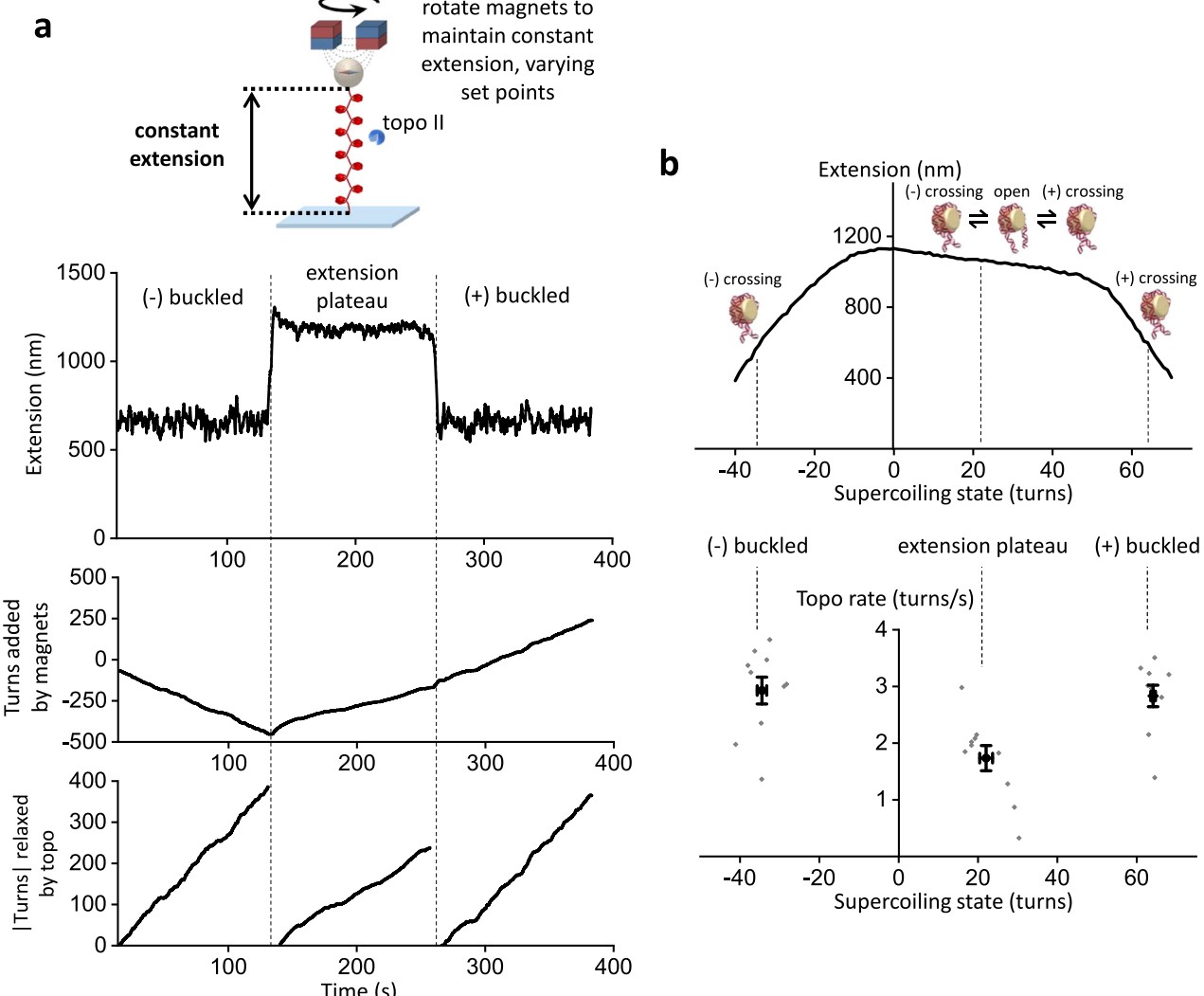

**Fig. 7 | Topo II relaxation of chromatin depends on compaction.**
**a** Measurements of topo II relaxation rate on chromatin fibers in different topological states. A chromatin fiber with an active topo II was held for 2 min in each of the following states: the (-) buckled state at 50% of its maximal extension, then at the midpoint of the extension plateau regime, then finally in the (+) buckled state at 50% of its maximal extension. The mean rate of topo II in each topological state was calculated from the number of turns relaxed by the bound topo II in the step corresponding to that topological state. **b** Topo II rate on chromatin fibers as a function of supercoiling state ($N = 10$ biologically independent traces at each supercoiling state). Both individual data points (gray) as well as their means and SEMs for each topological state (black) are shown. For clarity, the chromatin ActA. Image in 7a is reprinted from ref. 21, with permission from Elsevier.

corresponding chirality, and an open state, in which the entry and exit DNA segments do not cross[43,46]. In the (+) and (-) buckled regimes of the chromatin extension-supercoiling state relationship, individual nucleosomes reside primarily in the corresponding crossed state. In comparison, pre-buckled chromatin in the extension plateau regime comprises a mixture of nucleosomes in the three states, with the partitioning determined by the degree of supercoiling. The extension plateau is a result of the gradual conversion of nucleosomes from the (-) state to the open and (+) states to absorb (+) supercoiling. Although the (+) and (-) states readily provide a crossing for topo II activity, the enzyme must rely on spontaneous looping to form a crossing for the open state, leading to slower topo II activity in the extension plateau regime compared to in the buckled regimes.

## Discussion

Our results highlight how the dynamics of supercoiled DNA and chromatin substrates impact topo II activity. Specifically, our data suggest that substrate dynamics dictate the processivity and rate of topo II activity by determining the availability of DNA crossings (Fig. 8).

We unexpectedly found that topo II relaxed buckled DNA with greater activity lifetime and processivity (Figs. 2 and 3) than previous estimates. Our data show that for buckled naked DNA, DNA crossings are readily available for topo II activity. In vivo, buckled DNA is formed by high levels of torsional stress, which can arise from transcription and replication. Our estimated lower bound of the processivity of a single topo II on buckled DNA far exceeds the necessary value to fully relax all of the supercoiling resulting from transcription of a typical gene (~1.5 kb[47]) or replication between adjacent origins (~30 kb[48,49]) in yeast. These processes result in ~150 or ~3000 turns of supercoiling, respectively, both of which are far less than the processivity measured by our experiments (>6000 turns). Thus, a topo II that is actively relaxing supercoiling from either process will almost certainly continue as long as the DNA substrate is buckled. During transcription, the high processivity on buckled DNA may enable topo II to counteract the increased transcription that occurs during gene activation and transcriptional bursting[50].

We found that topo II could relax naked pre-buckled DNA, although with reduced processivity compared to its activity on

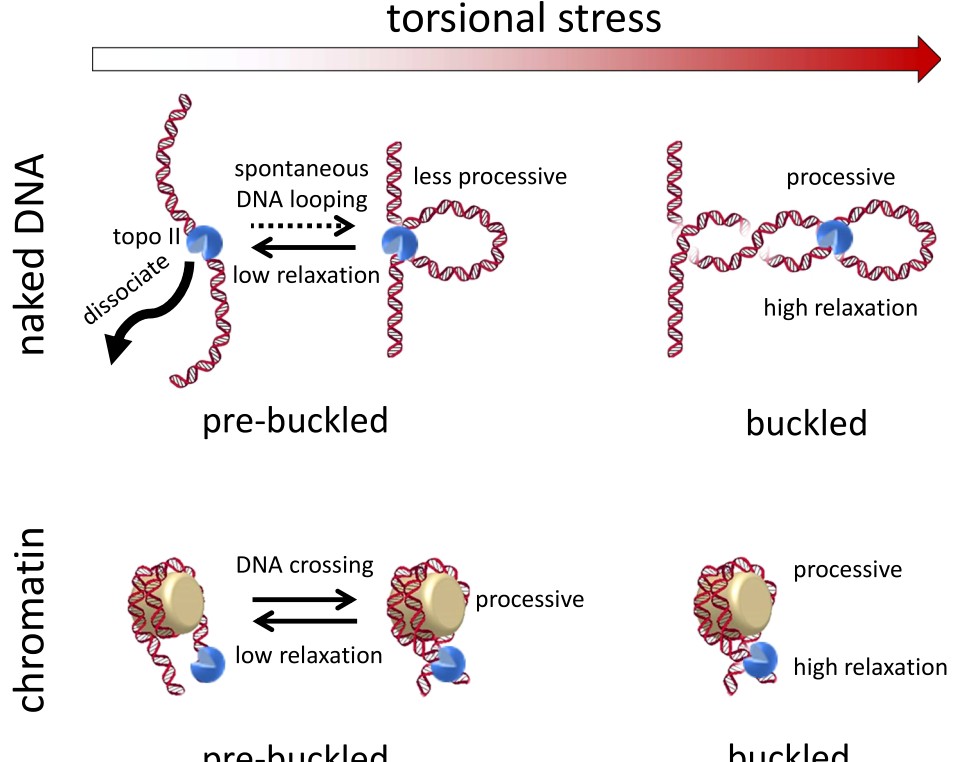

**Fig. 8 | Chromatinization modulates topoisomerase II processivity.** Topo II can capture spontaneous loops for activity on pre-buckled naked DNA, but with greatly reduced activity lifetime, processivity, and rate compared to activity on buckled naked DNA. Chromatinization enhances topo II activity lifetime and processivity, but the rate of activity on pre-buckled chromatin is reduced compared to that on buckled chromatin.

buckled DNA. Although previous single-molecule experiments were not sensitive to this capability, we detected and characterized topo II activity in the pre-buckled regime (Fig. 4). We show that a simple model relating this activity to the probability of spontaneous loop formation can predict our measurements, suggesting that topo II captures spontaneous loops to relax pre-buckled DNA.

Although previous studies have examined topo II activity on nucleosomal substrates[21,26,27], this work provides the first single-molecule examination of its dependence on the topological state of chromatin. Our data show that chromatinization greatly enhances topo activity lifetime and processivity (Figs. 5 and 6), possibly due to the intrinsic DNA crossings within nucleosomes. The persistent activity of topo II on chromatin may enable maintenance of super-helical density at levels near zero as a housekeeping function throughout the genome and during steady-state transcription. In addition, the increased activity lifetime of topo II may ensure that the enzyme remains bound to daughter chromatin fibers under low torsional stress to promote decatenation after replication termination. This inherent stability raises the possibility that the dissociation of topo II from chromatin may need to be assisted by motor proteins, such as RNA polymerases, replisomes, or chromatin remodeling factors.

We directly measured the rate of topo II relaxation on single chromatin fibers in different topological states (Fig. 7). Our results show that topo II relaxation rates are slower on pre-buckled chromatin and faster on buckled, more compact chromatin substrates. Taken with the chromatin extension-supercoiling state relation, this rate variation suggests a dependence on the conformations of individual nucleosomes within the fibers. In vivo, the relatively slow relaxation of (+) supercoiling in pre-buckled chromatin downstream of an active RNA polymerase may permit supercoiling-induced destabilization of nucleosomal barriers ahead of the polymerase, while the faster

relaxation of buckled chromatin by topo II may prevent transcription from stalling under high torsional stress[51–53].

Our work elucidates the dynamics of topo II relaxation of both naked DNA and chromatin substrates and provides insights into how topo II responds to the complex torsional profile of DNA that likely occurs within a cell. Although this study primarily focused on yeast topo II, we found that human topoisomerases IIα and IIβ also show exceptionally long activity lifetimes and processivities on buckled DNA (Fig. 3), indicating the topo II properties revealed in this study may be consistent for all type IIA topoisomerases. Future studies complementing the methods described in this work could further demonstrate the general importance of substrate dynamics in controlling fundamental biological processes.

## Methods
### DNA template construction
The 12.7 kb DNA construct with random sequence is composed of a 12,688 bp center segment flanked by 500 bp multi-labeled adapters at each end[32]. In brief, the center segment was PCR amplified from λ DNA using primers designed with 2−4 mismatched nucleotides to engineer restriction enzyme recognition sites (primer sequences 5'-CTACCC-GAGTGCGTGACAT-3' and 5'-CCAGTCTCGTGAAGCGGTA−3'). The resulting DNA was double digested with AvaI (NEB R0152) and BssSI-v2 (NEB R0680) and spin column purified to produce a center segment with overhangs for ligation. The multi-labeled adapters for the 12.7 kb DNA construct were 500 bp and PCR amplified from pNFRTC[21] with 24% of dATP replaced by biotin-14-dATP (Invitrogen 19524016) for the biotin adapter (primer sequences 5'- CAGTCACGAGGTTGTAAAACG-3' and 5'-ACGCCAAGCTTCCACATC−3') and 24% of dTTP replaced by digoxigenin-11-dUTP (MilliporeSigma 11093890) for the digox-igenin adapter (primer sequences 5'- GGGTAACGCTCGGGTTTTCC-3' and 5'-ACGCCAAGCTTCCACATC−3'). The labeled PCR products were

spin column purified, restriction digested with BssSI-v2 (for the biotin adapter) or AvaI (for the digoxigenin adapter), and ligated to the ends of the center segment. The sample was gel purified to remove unligated adapters.

The torsionally constrained 64-mer DNA construct is composed of a 12,667 bp center segment containing 64 tandem repeats of a sequence containing the 601 nucleosome positioning element (NPE)[21]. The center segment was formed by inserting the 64 tandem repeats into pUC19, double digesting, and then spin column purifying. It was then ligated to adapters generated in the same way as those used for the 12.7 kb DNA described above, but with primer sequences 5′- GTAAAACGACGGCCAGTG-3′ and 5′-ACGCCAAGCTTCCACATC −3′ for both adapters and digested with BglI (Thermo FD0074) for the biotin adapter or BstXI (Thermo FD1024) for the digoxigenin adapter.

The 6.5 kb DNA construct is composed of a 6481 bp center segment flanked by ~500 bp multi-labeled adapters at each end. First, λ DNA was digested with BamHI-HF (NEB R3136) and KpnI-HF (NEB R3142), after which the largest fragment (11,552 bp) was gel purified. This fragment was then inserted into pUC19 (NEB N3041) to form pL14225 (a.k.a. pMDW133). A 6546 bp sequence was then PCR amplified from this plasmid using LongAmp Taq DNA Polymerase and one primer designed with one mismatched nucleotide to engineer a restriction enzyme recognition site (primer sequences 5′- GCCACCTGACGTC-TAAGAAACCATTATTATCA-3′ and 5′-GCAGGTCCTTTCCGGCAAT-CAGG-3′, mismatch is underlined). The PCR product was spin column purified and then digested with BssSI-v2 and PpuMI (NEB R0506) in order to form the 6481 bp center segment with 4 or 3 bp overhangs on the ends, respectively. The ~500 bp multi-labeled adapters for the 6.5 kb DNA construct were created similarly to those for the 12.7 kb DNA construct. The adapters were PCR amplified from pNFRTC with Taq DNA polymerase and primers designed with 2 or 3 mismatches (forward primers 5′-GCTTCACTCGTGCTTTTGTTCCTTATTTT-3′ for the digoxigenin adapter and 5′-CGTTGTAAAAGGACCCCCAGTGAAT-3′ for the biotin adapter, mismatches are underlined; reverse primer 5′-ACGCCAAGCTTCCACATC-3′) with the same fraction of labeled dNTPs as the adapters for the 12.7 kb DNA construct. The PCR products were spin column purified and digested with BssSI-v2 or PpuMI before ligation to the 6.5 kb center segment. The sample was then gel purified before use.

## Protein purification

Yeast topoisomerase II, human topoisomerase IIα, and human topoisomerase IIβ were purified from yeast[54]. In brief, tagged topoisomerases were expressed in *S. cerevisiae*, which were harvested by centrifugation, flash frozen dropwise in liquid nitrogen, and lysed by cryogenic grinding. The tagged protein was purified from the lysate by a series of purification selection columns with the tags removed in an intermediate step. The purified proteins were filter-concentrated and flash-frozen for storage.

Histone octamers were purified from Hela-S3 cells[21,55,56]. In brief, nuclei were extracted from HeLa-S3 cell pellets (National Cell Culture Center HA.48), and core histones were purified using a hydroxyapatite Bio-gel HTP gel slurry (Bio-Rad Laboratories). The purified histones were stored at −80 °C.

## Torque measurements using the angular optical trap

The torque measurements were carried out with our in-house created AOT[29,30,57]. The AOT permits simultaneous measurement and control of force, extension, torque, and rotation of DNA via nanofabricated quartz cylinders[21,30,32,58,59]. A cylinder may be rotated via rotation of the trapping laser polarization, and the torque on the cylinder is directly measured via a change in the angular momentum of the laser after interaction with the cylinder. For the data shown in Fig. 1, the extension was filtered by a 0.05 s sliding window, and the torque was filtered by a 2 s sliding window. These experiments were performed in the topo

reaction buffer (10 mM Tris-HCl pH 8.0, 50 mM NaCl, 50 mM KCl, 3 mM MgCl$_2$, 0.1 mM EDTA, 1 mM DTT, 0.5 mM TCEP, 1 mM ATP, and 1.5 mg/ml β-casein).

## Sample chamber preparation

A sample chamber consists of a flow cell sandwiched between two microscope coverslips coated with a thin layer of nitrocellulose by spin coating[21]. Prior to chamber assembly, coated coverslips were rinsed and stored in a clean plastic container for 48 h or more before being assembled into a microfluidic sample chamber using inert silicone high vacuum grease (Dow Corning 1597418). In contrast to previous work, nitrocellulose-coated coverslips were used for both the top and bottom surfaces of the sample chamber instead of only the bottom surface.

Tethers for MT experiments were formed in a sample chamber[21,22]. First, a nitrocellulose coated sample chamber was incubated with 10 ng/μL anti-digoxygenin (Vector Labs MB-7000) in phosphate buffered saline for 30 min. Next, 1 μm magnetic beads (Dynabeads MyOne Streptavidin T1, ThermoFisher 65601) coated with biotin and digoxigenin labeled DNA were bound to the surface to serve as fiducial markers. Then, 5 mg/ml β-casein (MilliporeSigma C6905) was incubated in the chamber for at least 1 h to passivate the surfaces. The DNA substrate of interest was then incubated at concentrations ranging from 1.5 pM to 10 pM for 15 min, followed by magnetic beads at a concentration of 20 μg/ml for 15 min. The buffer in the sample chamber was then exchanged for the topo reaction buffer.

## Magnetic tweezers

All experiments were carried out on a custom-built MT setup based on a previous design[21,22]. In brief, a pair of 0.25″ cube neodymium dipole magnets (K&J Magnetics B444) with a separation gap of 0.5 mm maintained by aluminum spacers was used to generate the magnetic field over the stage of a microscope. Magnetic bead images were collected via a 20x/0.75 NA air objective lens (Nikon Plan Apo 20 × 0.75 NA) onto a 2.3 MP camera (Basler acA01920-155 μm) at a frame rate of 40 fps and an exposure time of 0.75 ms. Magnetic bead positions were tracked in three dimensions with LabVIEW.

## Selection of a DNA tether with a single active topo II

Immediately prior to each topoisomerase experiment, 0.5 pM of topo II in topo reaction buffer was introduced into the chamber and incubated for 2 min at room temperature. The sample chamber was then flushed with 5x chamber volume of the topo reaction buffer without topo II, then flushed again after 5 min. These two washes reduce further topo II binding events during measurements. After this process, typically ~10–15% of our tethers showed topo II relaxation activity when wound down, which we defined as "active." Because of this low probability of a tether being initially active, the initial activity from these tethers was most likely from a single topo II molecule. However, during the course of the subsequent measurement, an additional topo II molecule, which could be a consequence of incomplete flushing or topo II unbound from other tethers, could bind to these tethers and relax them. To determine the likelihood of this occurrence, we performed the conventional "wind down and wait" control experiment[19] after topo II binding and flushing and monitored the tethers that were initially inactive but became active later in the measurements. We found this probability to be ~17 ± 5% during the entire 30 min experiment. For a shorter duration of 30 s (the pause threshold used to define the activity duration), this probability was only ~0.3 ± 0.1%. Therefore, our measurements should be predominantly from a single topo II. Furthermore, we found that the active tether fraction decreased from 70% to 10–15% with the two additional washing steps, indicating that the washes are effective at removing free topo II. In addition, experiments with the constant-extension method provided a natural method to select out traces with multiple topo II molecules, so

we were able to impose more stringent selection criteria to ensure a single topo II enzyme condition. In this method, the onset of activity from an additional topo II molecule caused an abrupt doubling in the relaxation rate. Thus, we could readily identify and exclude those traces from further analysis. All experiments were performed at 23 °C.

For experiments that required the selection of one naked DNA tether with an active topo II, tethers were initially wound down by adding +60 turns by magnet rotation at 10 turns/s. 10 s after winding, a tether was chosen randomly from the tethers with extension increase greater than 250 nm. For experiments that required the selection of one chromatin fiber with an active topo II, tether selection was performed similarly but with the tethers initially wound down by adding −30 turns instead. In this way, we limited possible structural deformation of nucleosomes that could be induced by (+) torsion[60].

### Implementation of the extension clamp

Supplementary Fig. 1 explains the overall operation of the extension clamp. We show a schematic flow chart of the extension feedback control and algorithm (Supplementary Fig. 1b). The feedback control was implemented as a proportional controller in LabVIEW. Given a setpoint extension, the error between the setpoint and the extension of a selected tether was calculated for each frame. Every five frames, the mean of the error was calculated for the previous five frames then multiplied by the gain of the proportional controller (0.05 turns/s/nm for all experiments) in order to set the magnet rotation rate. We also show representative raw traces of how the clamp responds to bead position changes (Supplementary Fig. 1c, d) and validation of the method by the agreement of the topo II relaxation rate of post-buckled DNA measured with the constant-extension method and the conventional method (Supplementary Fig. 1e).

### Pre-buckled supercoiling state calculation

In the repeated winding experiments to probe topo II activity on pre-buckled DNA (Figs. 2 and 4; Supplementary Fig. 2), the DNA was buckled immediately after each winding step. The measured extension at this time point was used to determine DNA supercoiling state in the preceding step (as in Supplementary Fig. 2a). To do so, we used the mean extension over 1 s after rewinding and the extension-supercoiling state relation to determine the supercoiling state after the rewinding step, subtracted it from the number of turns added in the rewinding step, and added a small number of turns to account for topo II activity on buckled DNA during the 0.875 s rewinding step and 1 s averaging after rewinding at the measured rate (described below in the next section). The resulting value provides the supercoiling-state of the DNA in the proceeding step.

### Topo II rate calculation from repeated winding experiment

In the repeated winding experiment used to measure topo II activity on pre-buckled DNA (Fig. 4), the rate of topo II activity on buckled DNA was calculated while the DNA was still buckled after each winding step (before relaxation to the buckling transition). The extension data in this regime was first filtered by a 5 s sliding window and converted to supercoiling state by using the initially measured extension-supercoiling state relation. The rate was then calculated for each relaxation step from linear fits to the supercoiling state data. The mean of such rates calculated from each force condition was used for the corresponding linear extrapolation of buckled topo II rates on pre-buckled DNA (Fig. 4, dashed blue lines).

### Conversion of pre-buckled extension to supercoiling state

For the direct conversion of the measured extension to supercoiling state in the repeated winding experiment used to characterize relaxation of pre-buckled DNA by topo II (Fig. 4b), a mean extension versus time curve was obtained by averaging all relaxation traces after alignment at the buckling transition. The supercoiling state versus time curve was then determined from this extension versus time curve using the extension-supercoiling state relation at this force.

### Coarse-grained Monte Carlo simulations of DNA

The DNA was modeled as a discrete worm-like chain. Equilibrium configurations of naked DNA under external force and torsion were then generated using Monte Carlo (MC) simulations with a Metropolis-Hastings procedure[61], with parameters shown in Table S1. As the model does not directly incorporate the twisting of individual chain segments, the overall twist change $\Delta Tw$ in the DNA was indirectly computed by computing the global writhe change $\Delta Wr$ of the chain[62] and invoking the conservation of linking number $\Delta Lk = \Delta Tw + \Delta Wr$.

Using a scheme described in a recent study[63], we implemented two types of structural moves to effectively sample a wide range of DNA configurations as part of the Metropolis process: local chain moves (pivot and crankshaft rotations) and full chain exchanges between neighboring replicas. For the local moves, we sampled the degree of pivot and crankshaft rotations from a uniform distribution between [−50°,50°]. To implement the full chain exchanges, 12 parallel simulations were simultaneously generated on separate CPU cores. MC samples were then generated by applying 1000 local move trials followed by replica-exchange trials between configurations with neighboring linking numbers, $\Delta Lk_i$ and $\Delta Lk_{i+1}$, whose modified Metropolis criterion is given by the differential energy term:

$$\Delta E = \frac{4\pi^2 L_t}{\beta L_c}\left(\Delta Lk_i - \Delta Lk_{i+1}\right)\left(\Delta Wr_i - \Delta Wr_{i+1}\right) \quad (1)$$

where $1/\beta$ is the thermal energy, $L_t = 109$ nm is the twist persistence length, and $L_c$ is the contour length of the DNA (Table S1). The configurations were saved after five successful replica-exchange trials between any pair of neighboring simulations, and the process was repeated until there were 25,000 saved independent configurations for each simulation.

Since MC simulations allow for segments to cross one another, such moves could lead to creation of knots. Hence, a knot-checking algorithm[64] was implemented after every 1000 moves, and any conformation that contained a knot was rejected. To facilitate equilibration of long DNA molecules, we parallelized the writhe computation algorithm[62] and used MATLAB Parallel Computing Toolbox to increase the overall sampling speed. To confirm that thermal equilibrium was achieved, we plotted the simulated DNA extension-supercoiling state relations and compared them to the experimentally measured extension-supercoiling state relations. Once the equilibriums were attained, subsequent saved DNA configurations were analyzed to estimate the average rate of DNA crossing formation.

We obtained the kinetics of DNA crossing formation from our saved DNA conformations. First, we computed the distribution of all site-to-site distance measurements $R$ between every single pair of non-contiguous vertices of all our saved conformations. Next, the distributions were transformed into effective free energy profiles that were used to model one-dimensional diffusion along the distance reaction coordinate $R$ and estimate the average time it takes to reach from an "uncrossed" state to a state with a crossing. In our analysis, a crossing (sometimes called a DNA juxtaposition[65]) is defined as a state where two segments of the same DNA molecule are within a maximal threshold $R_T = 10$ nm distance of each other. Finally, the mean first passage time $\langle T \rangle$ for crossing formation for a given superhelical density σ and force $F$ can be estimated using:

$$\langle T \rangle_{\sigma,F} = \frac{1}{DZ}\int_{R_T}^{L} dR \int_{R_T}^{R} dR' \int_{R'}^{L} dR'' e^{-\beta(F_{\sigma,F}(R) - F_{\sigma,F}(R') + F_{\sigma,F}(R''))} \quad (2)$$

where $Z = \int_{R_T}^{L} dR e^{-\beta F_{\sigma,F}(R)}$ is the uncrossed state partition function, $F_{\sigma,F}(R)$ is the effective free energy profile along the reaction coordinate

$R$ for a given set of $\sigma$ and $F$, and $D$ is the effective diffusion coefficient. We then used the average rate of DNA crossing $k_{\sigma,F} = \frac{1}{\langle T \rangle_{\sigma,F}}$ (Supplementary Fig. 6) to model the supercoiling relaxation activity of topo II by using a Michaelis-Menten type equation to model the rate of supercoiling relaxation by topo II:

$$\frac{d\Delta Lk}{dt} = V_{max} \frac{k_{\sigma,F}}{k_{\sigma,F} + k_{1/2}} \tag{3}$$

where $V_{max}$ and $k_{1/2}$ are constants that could be obtained by comparing to experimental data. Specifically, $V_{max}$ was obtained experimentally from the post-buckling relaxation rate in the repeated winding experiment (3.4, 3.1, and 3.0 turns/s for 0.5, 1.1, and 1.6 pN, respectively) and $k_{1/2} = 6.8 \pm 1.3$ loops/s was a global fitting parameter computed by integrating the theoretical relaxation rate over time and minimizing the chi-squared difference between the integrated theoretical curves and experimental data (Fig. 4b).

To examine how DNA bending by topo II may affect the DNA crossing rate, we repeated the simulations at $F = 0.5$ pN with the introduction of a rigid bend in the center of the DNA (Supplementary Fig. 7a). The geometry of this bend structure is based on a two-kink model[39], where the DNA is kinked in two positions, creating a flat-bottomed 'V' with a total bend angle of 120°. The average looping rate was calculated in the same way as before (Supplementary Fig. 7b), and the results were fitted to the same Michaelis-Menten type equation. We found that the simulations including a DNA bend could be consistent with our results, but with $k_{1/2}$ increased to $14.6 \pm 3.9$ loops/s. The greater value of $k_{1/2}$ reflects the increased formation of DNA loops at low values of superhelical density, allowing for more efficient loop capture by topo II to match our experimental results (Supplementary Fig. 7c).

## Nucleosome assembly

Nucleosome arrays were assembled onto the 64-mer DNA construct[21]. In brief, histones were deposited onto the 64-mer DNA by gradient salt dialysis from 2.0 M to 0.6 M over 18 h at 4 °C at different molar ratios (0.25:1 to 2.5:1) of histone octamers to 601 DNA repeats[66,67]. An equal mass of 147 bp random-sequence competitor DNA was included to avoid nucleosome over-assembly[21]. The quality and saturation level of the final product were assayed by gel electrophoresis in 0.7% native agarose and then by stretching with an optical tweezers setup.

## Chromatin fiber characterization on MT

Prior to each topoisomerase experiment on a chromatin substrate, we performed a twisting experiment[21] to measure the extension-supercoiling state relation of the substrate under a constant force of 0.5 pN in the range −40 to +70 turns (Figs. 5a and 7b). These extension-supercoiling state relations were fit to five-piece functions composed of three linear regions joined by two parabolic regimes (Supplementary Fig. 8a)[21]. From these fits, we obtained the peak extension and the turns of the left and right "buckling-like" transitions $w_t^-$ and $w_t^+$, which were defined to be the intercepts of the middle linear regime with the left and right linear regimes, respectively[21]. We compared these parameters with those of chromatin fibers that were well-saturated with primarily full nucleosomes (as opposed to subnucleosomal structures, such as tetrasomes), which we previously characterized with the nucleosome disruption assay described below. This enabled selection of chromatin fibers with acceptable saturation and composition using only the initial extension-supercoiling state relation, without disrupting the nucleosomes.

To benchmark the nucleosome saturation and composition of a chromatin substrate, we performed a separate nucleosome disruption assay. After measuring initial extension-supercoiling state relations as described above, we disrupted the nucleosomes in the fiber

with a combination of high stretching force and high ionic strength and interpreted the resulting extensions based on a previously-established multi-stage disruption model of chromatin[21,42,66–69]. Specifically, we examined the extension of each chromatin fiber at 2 pN, at which the "outer-turn" DNA (interacting primarily with the H2A/H2B dimers) remains bound to the histone core octamers, and 6 pN, at which the outer-turn DNA has been mostly released. The outer-turn DNA is ~72 bp in length for each nucleosome, so the number of nucleosomes initially containing a wrapped outer-turn $N_{out}$ could be determined from the DNA length released. Next, we fully dissociated nucleosomes with a 5 min incubation in 2 M NaCl followed by an additional wash with the same salt concentration. This process released the "inner-turn" DNA (interacting primarily with the H3/H4 tetramer) from the histone core octamers, which is ~75 bp in length for each nucleosome. The number of bound tetramers $N_{in}$ could be determined from the total DNA released by the high salt wash, and the length of the underlying naked DNA template could be measured after all nucleosomes were fully dissociated. These experiments were performed using nucleosome array samples assembled with varying molar ratios of histone octamers to 601 DNA repeats to obtain data over a broad range of saturations.

To establish benchmark relationships between nucleosome saturation/composition and the shape of the extension-supercoiling state relation using the data from the nucleosome disruption assay, we first considered the relative values of $N_{out}$ and $N_{in}$. We only considered fibers satisfying $(|N_{in} - N_{out}|/N_{in}) \leq 0.15$, which indicates $N_{out}$ and $N_{in}$ being comparable within measurement uncertainties, as having an acceptable composition of nucleosomes. For these fibers, we estimated the total number of full nucleosomes $N_{nuc} = (N_{out} + N_{in})/2$ and plotted the relationship between the peak extensions from their initial extension-supercoiling state relations and $N_{nuc}$ (Supplementary Fig. 8b, data points). A linear fit to this relationship (Supplementary Fig. 8b, dashed black line) was later used to calculate $N_{nuc}$ for each fiber prior to experiments with topo II. In addition, we plotted the relationship between the positions of the buckling-like transitions $w_t^\pm$ and $N_{nuc}$ for nucleosome fibers with acceptable composition (Supplementary Fig. 8c, data points). Linear fits to the relationships between $w_t^\pm$ and $N_{nuc}$ (Supplementary Fig. 8c, dashed black lines) were later used to evaluate the composition of nucleosomes in each fiber prior to experiments with topo II.

For experiments with topo II, we only selected fibers with maximal extension consistent with $50 \pm 5$ nucleosomes and with $w_t^+$ and $w_t^-$ within the 95% confidence intervals of the linear fits to $w_t^+$ and $w_t^-$ versus $N_{nuc}$ (Supplementary Fig. 8c, green regions). These selection criteria removed tethers with extreme nucleosome saturation, poor nucleosome composition, short underlying DNA (i.e., loss of repeats in the underlying 64-mer DNA), or partial sticking of the fiber to a surface. For the fibers that passed selection with ~50 nucleosomes, $w_t^+ \approx 50$ turns, suggesting that each nucleosome absorbs about 1 turn before buckling.

## Reporting summary

Further information on research design is available in the Nature Portfolio Reporting Summary linked to this article.

## Data availability

Source Data are provided with this paper. The Source Data File contains all data shown in the main text and Supplementary Figures. All other data that support the findings of this study are available from the corresponding author upon request. Source data are provided with this paper.

## Code availability

All custom MATLAB code for data analysis in this work is available at https://github.com/WangLabCornell/NCOMMS-23-05291A[70].

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

## Acknowledgements

We thank members of the Wang Laboratory and Drs. Z.J. Wang and G. Lambert for helpful discussion and comments. We especially thank Drs T.T. Le and X. Gao for technical advice and generation of the naked DNA templates used in this study and Dr. J.P. Sethna for technical advice regarding the Monte Carlo simulations in this study. This work is supported by the National Institutes of Health grants R01GM136894 (to M.D.W.), T32GM008267 (to M.D.W.), and R01-CA077373 and R35-CA263778 (to J.M.B.). M.D.W. is a Howard Hughes Medical Institute investigator. Figures 1a, 2a, b, 3a, 4a and Supplementary Figs. 1a, 3a contain images modified from those from RNA Polymerase as a Torsional Motor. J.L. Killian, J. Ma, and M.D. Wang. Chapter 3: RNA Polymerases as Molecular Motors: On the Road (2022)[28], with permission from the Royal Society of Chemistry. Figures 5a, b, 6a, 7a contain images reprinted from Cell, 179, T.T., Gao, X., Park, S., Lee, J., Inman, J.T., Lee, J.H., Killian, J.L., Badman, R.P., Berger, J.M., and Wang, M.D. Synergistic Coordination of Chromatin Torsional Mechanics and Topoisomerase Activity, 619-631, Copyright (2019)[21], with permission from Elsevier.

## Author contributions

J.L., J.T.I. and M.D.W. designed single-molecule assays. J.L., M.W., J.T.I. and Y.H. performed experiments. G.S. performed simulations. J.L., M.W. and Y.H. analyzed data. J.T.I. constructed the MT with contribution from J.L. on software. J.H.L., J.J. and J.M.B. purified topo II and characterized it biochemically. R.M.F. purified histones. S.h.P. and M.W. prepared and characterized the chromatinized DNA substrates. J.L., G.S. and M.D.W. wrote the initial draft, and all authors contributed to manuscript revisions. M.D.W. supervised the project.

## Competing interests

The authors declare no competing interests.
