## [Peer Review File · Nature Communications]

Chromatinization Modulates Topoisomerase II ProcessivityREVIEWER COMMENTS

Reviewer #1 (Remarks to the Author):

In their manuscript, Lee and coauthors present exciting new single-molecule data on the activity of yeast topoisomerase II. They use cleverly designed and highly innovative new magnetic tweezers assays to assess topoisomerase activity on different substates (naked DNA, chromatin) with well-defined supercoiling / plectonemic states. Using this approach, some key and surprising new findings have been made: the processivity of topoII on plectonemic DNA is much larger than thought before. Furthermore, they provide the first quantitative assessment of topoII activity on "pre-buckled" DNA and show that topoII binds substantially shorter to this DNA and has a 2 orders of magnitude lower processivity. Using further experiments and Monte Carlo simulations they show that this difference can be understood in terms of DNA loops being required for topoII activity: DNA crossings or loops are always there in plectonemes and only intermittently, driven by thermal fluctuations in the pre-buckled state. Using a chromatin model, they show that chromatin, which has plenty DNA crossings, results in an increased topoII activity.

Taken together, this is a very nice manuscript: innovative new approaches, great, impactful and important findings. Furthermore it is very well written and accessible. I fully support publication in Nature Communications. Content-wise I do not have questions / remarks. The authors could consider looking into the two textual suggestions below, which might help to make the manuscript even more accessible.

1) 2nd paragraph of the results section. The authors conclude that "a single topoII molecule" remained active for 30 minutes. Here they do not explicitly explain why this conclusion can be drawn here. Later, for other experiment, they do discuss this. It might be helpful to also (or instead) discuss here why this statement can be made.

2) "TopoII is less processive on ..." section. End of 2nd paragraph. I could not completely understand the sentences "The extension after each of those ... due to the short duration of the rewinding steps". Could the authors try to expand this discussion a little bit? The assay introduced here is superelegant, but also somewhat complex. A bit more explanation here, could help the general reader.

Reviewer #2 (Remarks to the Author):

Lee and co-workers use magnetic tweezers to study the yeast topoisomerase II relaxation of positively supercoiled DNA in the pre- and post- buckled state along with the relaxation of positively and negatively supercoiled chromatinized DNA substrates.

The authors observe a highly processive relaxation of positively supercoiled DNA in the post-buckled state while a lower processivity and lower binding time in the pre-buckled state. The relaxation in the pre-buckled state is attributed to the capture of spontaneous loops that the topoisomerase can capture.

On chromatinized DNA the authors observe enhanced binding and relaxation processivity at low supercoiling and find that there are subtle differences in relaxation rate in three distinct phases of chromatin.

Overall these results provide some new insights into the activity of yeast type II topoisomerase – principally its high processivity, however the main results are largely incremental in nature as they modestly extend and largely confirm what is known about the mechanism of type II topoisomerases. This work is much better suited for a more specialized journal as laid out in detail below.

- What are the noteworthy results?

This study slightly extends previous measurements of binding and relaxation activities of topo II in the prebuckling region of DNA supercoiling, particularly in the presence of chromatin (1) (2, 3), but largely confirms previous findings from many different methods that topo II preferentially binds to and relaxes DNA crossings (4-6).

- Will the work be of significance to the field and related fields? How does it compare to the established literature? If the work is not original, please provide relevant references.

Although the work was not sufficient to fulfill the objective of this study to address the effect of DNA torsion on topo II activity, the work presents some very specific and detailed findings that could be of interest to the fields of DNA topology and topoisomerases.

Specifically:

The authors claimed that there was no direct evidence for supercoil relaxation prior buckling by topo II.

“but how topo II activity responses to torsional stress is not fully understood. Previous biochemical and single-molecule studies have successfully elucidated many aspects of topo II activity on naked plectonemic DNA^{6, 15, 16, 17, 18, 19, 20}. However, torsion can also accumulate in DNA prior to plectoneme formation, and there has yet been no direct evidence for topo II relaxation of such a substrate. More importantly, even less is known about the processivity and rate of topo II relaxation of chromatin, the primary substrate of topo II in vivo.”

However, the subject has been addressed by Strick et al using *D. melanogaster* topo IIA who also provided the same conclusion that efficient topo IIA supercoil relaxation requires presence of DNA crossings or DNA writhe and that the relaxation rates would be limited prior to plectoneme formation (2) (p.606-607). Additionally, the same article showed the DNA crossover capture by topo II prior to buckling transition based on the elegant single molecule experiment and similarly for topo IV (7).

Investigating the relaxation rate and the binding stability of topo II prior to a buckling transition might be good to understand the effect of chromatin on DNA topological state and geometry and the subsequent topo II activity. However, it is not sufficient to understand torsional effects on topo II supercoil relaxation activity. For topologically constrained DNA, DNA torsion increases linearly as increasing twist (T_w) in terms of the bead rotation prior to plectoneme formation (buckling transition) and this stored torsion will remain roughly constant after the buckling transition. Thus, it seems inadequate to study the effect of torsion on topo II activity prior to buckling transition as torsion would vary dependent on the residual twist while topo II removes supercoils. To address the question posed regarding the effect of DNA torsion on the activity of type II topoisomerase would require a systematic study of the rate and processivity of relaxation under differing levels of well-controlled constant torsion, which has been done previously for other type II topoisomerases. The current work explores only a single value of positive torsion in the supercoiling regime and poorly defined positive torsion in the pre-buckling regime.

In order to understand how topo II reacts to a different level of torsion of supercoils, I would recommend the authors to perform under a range of different torques for both positive and negative supercoils as done for other topoisomerase studies (3, 8) since supercoiling at different tension with magnetic tweezers leads to a different torsion for plectoneme DNA. Only torsion not tension would apply for topo II activity if topo II binds to DNA plectoneme during relaxation.

If the authors are keen on understanding how crossing probability affects topo II enzyme rate, I would recommend the authors to employ the single molecule braiding technique that was pioneered by the Bensimon-Croquette group (9).

- Are there any flaws in the data analysis, interpretation and conclusions? - Do these prohibit publication or require revision?

Binding time of topo II for prebuckled region estimation is acceptable despite probing with periodic magnet turns, but the processivity estimate obtained from this approach does not reflect the relaxation processivity of prebuckled DNA by topo II as the measured processivity was from the relaxation of 30 turn that encompass supercoils for both pre and post buckling transition.

Furthermore, it wasn't the enzyme that stopped relaxation during the waiting time between turns. Put simply the approach can measure the binding time in the pre-buckled state but cannot accurately provide a measure of the processivity in this state.

I strongly recommend the authors should perform binding time measurements on negatively

supercoiled DNA prior and post buckling transitions in order to compare the binding time of negatively supercoiled chromatin DNA data.

- Is the methodology sound? Does the work meet the expected standards in your field?

The constant extension technique could be an ideal way to maintain the level of supercoiling substrate to be constant for topoisomerase if the spatial and temporal resolutions are higher than the enzymatic activity. If not, the error of enzymatic rate estimation that was obtained from the turn data in the constant-extension would be couple to the extension measurement error per each turn due to the limited spatial and temporal resolutions. I would recommend the authors to validate the constant extension technique by comparing the rates from the constant torque method where a constant number of turns will be applied when the extension reaches 90% relaxation of plectoneme.

- Is there enough detail provided in the methods for the work to be reproduced?

The descriptions of the methods were adequate.

Detailed comments and concerns

Page 4.

"In the present work, we developed novel single-molecule assays that enable the direct..

... These experiments demonstrate the impact of varying degrees of torsion on topo II activity on plectonemic, non-plectonemic, and chromatinized DNA."

Firsthand, it seems inappropriate to consider the employed single-molecule assay to be "novel" as many similar magnetic tweezers based single molecule assays have been performed. This is a small extension over existing feedback approaches that maintain the DNA under a constant level of torsion in the plectoneme regime. Also I do not agree that the work showed varying degrees of torsion on topo II activity, as described in detail above.

Page 5.

"Unexpectedly, we discovered that topo II was much more processive on plectonemic DNA than previously thought.."

This claim requires a reference instead of indicating "previously thought". This claim is made repeatedly throughout the manuscript but is never substantiated by a citation.

Page 8.

The authors make the following disingenuous claim in which the processivity of yeast topo II measured in the current work is compared to the processivity of other type II topoisomerases as measured in the cited work.

"This lower bound on the processivity far exceeds values suggested by other single-molecule methods^{15, 19, 20}, indicating that topo II can be more processive on plectonemic DNA than previously thought."

The difference in processivity is a difference among enzymes rather than a difference among measurements, but the authors are confusing these two factors in a way that could be misleading. They would do better to state that the yeast enzyme is more processive than other type II topoisomerases, which is a valid statement that is one of the clear results of this work.

Page 9.

The authors state:

"Taken together, our findings establish that topo II is much more stable on plectonemic DNA (high torsional stress) compared to on non-plectonemic DNA (low torsional stress)."

This is incorrect. The torsional stress in the DNA is related to the twist whereas the formation of plectonemes is related to writhe in the DNA. In general this statement is completely incorrect and misleading. Under the same applied force, the torsion in the non-plectoneme regime is less than or equal to the torsion in the plectoneme regime, but without specifying this condition the authors have made an incorrect and misleading statement.

Page 10.

"used to precisely calculate the turn state from the hat curve.. precisely probe.."

I would recommend to remove “precisely” and also explain how precisely the authors calculated the turn state from the hat curve.

Page 16.

the authors make the following statement:

“We unexpectedly found that topo II relaxed plectonemic DNA with much greater stability and processivity than previously thought (Fig. 1). These values were not measurable with previous single-molecule techniques.”

This is inaccurate as similar feedback approaches have been used to measure highly processive relaxation event by type II topoisomerases (10).

Page 25.

I would recommend the authors to show some raw traces for implementing the constant-extension clamp.

Page 28 and 37 (figure 2C).

It is unclear based on the method what the two fitting parameters are. Please indicate in the main text what they are and their error bars.

Furthermore – the MC simulations ignore the effect of the type II topoisomerase bound to the DNA in inducing the bending and buckling of the DNA that is well established in the literature (7). Given that the type II topoisomerase is thought to remain bound in the pre buckled state, it will contribute significant bending energy leading to increased rate and probability of buckling in this state. This fundamental contribution of the binding of the type II topo to the DNA in driving the spontaneous looping was completely ignored in the simulations and in the related discussion.

Figure 1D. As stated previously, the measurement of the pre-buckled relaxation activity can not provide a measure of the processivity of the enzyme in this regime.

Figure 3. There is no caption for part D.

Figure S6

Adjust the x and y axis ranges to >40 turns in order to make data more visible.

In summary, the authors have measured the rate and processivity of yeast topo II on positively supercoiled DNA in the pre-buckled and plectoneme states. They also measure the rate and relaxation of chromatinized DNA under both negative and two distinct phases of positive supercoiling conditions. Their conclusions summarized in figure 5 are that yeast topo II remains bound longer, and is more active, on DNA molecules containing crossings. These results largely confirm what is known about type II topoisomerases, but extend the field to provide detailed rates and processivity measurements for yeast topo II, which has not been extensively studied with similar techniques. Although these measurements of yeast topo II activity will be of utility and interest to the community, the results do not provide significant broader insights or mechanistic details concerning type II activity to warrant publication in Nature Communications.

Literature cited:

1. Le TT, Gao X, Park Sh, Lee J, Inman JT, Lee JH, Killian JL, Badman RP, Berger JM, Wang MD. Synergistic Coordination of Chromatin Torsional Mechanics and Topoisomerase Activity. *Cell*. 2019;179(3):619-31.e15. doi: <https://doi.org/10.1016/j.cell.2019.09.034>.
2. Strick TR, Charvin G, Dekker NH, Allemand J-F, Bensimon D, Croquette V. Tracking enzymatic steps of DNA topoisomerases using single-molecule micromanipulation. *Comptes Rendus Physique*. 2002;3(5):595-618. doi: [https://doi.org/10.1016/S1631-0705\(02\)01347-6](https://doi.org/10.1016/S1631-0705(02)01347-6).
3. Strick TR, Croquette V, Bensimon D. Single-molecule analysis of DNA uncoiling by a type II topoisomerase. *Nature*. 2000;404(6780):901-4.
4. Roca J, Berger JM, Wang JC. On the simultaneous binding of eukaryotic DNA topoisomerase II to a pair of double-stranded DNA helices. *J Biol Chem*. 1993;268(19):14250-5. doi: [https://doi.org/10.1016/S0021-9258\(19\)85234-1](https://doi.org/10.1016/S0021-9258(19)85234-1).

5. Roca J, Wang JC. The probabilities of supercoil removal and decatenation by yeast DNA topoisomerase II. *Genes to Cells*. 1996;1(1):17-27. doi: 10.1046/j.1365-2443.1996.01001.x.
6. Zechiedrich EL, Osheroff N. Eukaryotic topoisomerases recognize nucleic acid topology by preferentially interacting with DNA crossovers. *The EMBO Journal*. 1990;9(13):4555-62. doi: <https://doi.org/10.1002/j.1460-2075.1990.tb07908.x>.
7. Charvin G, Strick TR, Bensimon D, Croquette V. Topoisomerase IV bends and overtwists DNA upon binding. *Biophys J*. 2005;89(1):384-92. Epub 2005/05/03. doi: 10.1529/biophysj.105.060202. PubMed PMID: 15863484; PMCID: PMC1366538.
8. McKie SJ, Desai PR, Seol Y, Allen AMB, Maxwell A, Neuman KC. Topoisomerase VI is a chirally-selective, preferential DNA decatenase. *eLife*. 2022;11:e67021. doi: 10.7554/eLife.67021.
9. Neuman KC, Charvin G, Bensimon D, Croquette V. Mechanisms of chiral discrimination by topoisomerase IV. *Proc Natl Acad Sci USA*. 2009;106(17):6986-91. doi: 10.1073/pnas.0900574106.
10. Seol Y, Gentry AC, Osheroff N, Neuman KC. Chiral discrimination and writhe-dependent relaxation mechanism of human topoisomerase II α . *J Biol Chem*. 2013 May 10;288(19):13695-703. doi: 10.1074/jbc.M112.444745. Epub 2013 Mar 18. PMID: 23508957; PMCID: PMC3650406.

Reviewer #3 (Remarks to the Author):

Lee et al. *Nature Communications* 2023/03

This is an interesting manuscript where a new single molecule technique was devised to enable continuous monitoring of yeast topo II relaxation to test how the enzyme responds to differing degrees of torsional stress. The new technique allowed the researchers to hold the winding of DNA constant (by spinning it as the enzyme relaxes it). This is as opposed to previous work where the enzyme relaxes plasmid (with a limited number of supercoils) and slows down as the plasmids become less supercoiled, thus giving different (and unmeasurable) rates over the different supercoiling ranges. The researchers go on to assess what seems to be a different DNA with chromatin. There are interesting results in the manuscript, but some conclusions are not supported by the evidence, previous work relevant to the new results is ignored, the manuscript is written for experts in single molecule experiments, experimental procedures are not well explained, and the authors use imprecise terminology.

Specifically:

1. Many previous publications revealed a strong dependence of topo II (from various organisms) for supercoiled DNA over relaxed DNA. And some have demonstrated a strong preference for one supercoiling handedness (+ versus -) over another. Yet all this previous work is ignored. The researchers must fit their new results into the known work critically.
2. The question of the generality of the observations is important and is too superficially dealt with by the authors in the manuscript discussion, "As all type IIA topoisomerases share the same core mechanism, the properties of yeast topo II revealed by methods used in this study may be consistent in all type IIA topoisomerases, including those in higher-order organisms." This question is so easily addressed by assessing different (preferably "higher order") topo II. Importantly, many studies (ignored in the manuscript) have comprehensively compared activities, including relaxation, by various topo II's and the differences across species are striking. To which enzyme, then, the researchers compare their new rates is not clear.
3. In the abstract, what is novel about the single molecule set-up should be mentioned. The experimental setup, which is novel and crucial to the study, should be explained clearly in the text. The diagram of the setup in Figure 1a is not helpful as it looks like any other magnetic tweezers experiment. How is the constant extension maintained? A schematic of the extension correction protocol (as part of figure 1) is needed. The description of the constant extension clamp is impossible to follow in the Methods.
4. The abstract would be improved by more precision in the measures. How much "faster"? The data generated are highly quantitative, so the abstract, indeed the entire manuscript, should be as well.

5. The authors use a term "pre-buckling" (and "pre-buckled") (to contrast with "plectonemic"). But the comparisons are not clear. Further, well defined terms apply to the situation. It seems that the authors are describing supercoiling that is only twist (pre-buckling) versus supercoiling that is only writhe (buckling? = plectoneme?). I would not introduce or use these less precise terms to describe what there are already terms for. If I am wrong (that "pre-buckling" = twist and "buckling" = plectoneme = writhe), then a much better explanation is needed to help readers follow. It is especially confusing terminology because "pre-buckled" could mean that the buckle was put in before the experiment (this is how I read it the first time through the manuscript). Or it could mean that it has not yet "buckled," which must be what is meant. But few readers will work as hard as I did to understand and follow. The loose term becomes even more problematic when used in the context of nucleosome binding. The sentence "Each nucleosome contributes about +1.0 turns to the (+) width of the hat tilt, implying that each nucleosome can absorb about 1.0 turns of (+) supercoiling before buckling" is read that the nucleosome buckles. Is that what was meant? Later, in the Methods section, a new term "buckling-like" is introduced. Because "buckling" is already a fuzzy term, what is meant by this term is even fuzzier.

6. The problem is that in these studies, the only read out is the length of the DNA and the rapidity of the rotor. Forthright acknowledgement of the limitations of the experimental set-up would be helpful for the non-experts.

7. The argument that topo II can relax "pre-buckled" DNA is not supported by data. Unless the authors provide more convincing evidence for this assertion, this claim should be dropped. To detect relaxation the researchers needed to rotate the magnets "to bring the molecules slightly into the plectonemic regime". The most likely explanation is that the topoisomerase is acting on the plectonemes formed during the "brief activity check". Not surprisingly the average processivity (~30 turns) corresponds to the number of turns added during the "brief activity check". The authors imply that the topoisomerase is acting on the non-plectonemic region of the DNA molecule. More likely explanations are that either the topoisomerase briefly dissociates and rebinds to a crossover, or the plectoneme is formed at the place on the DNA where the topoisomerase is bound. Yeast topoisomerase II imparts a significant bend in the DNA. This bending likely facilitates the formation of the plectoneme. If the newly introduced plectoneme is formed at the site where topoisomerase is bound, the enzyme should act very efficiently. The authors speculate that topo II is capturing spontaneous loops that form due to thermal fluctuation. While this explanation is feasible, it seems more likely that the enzyme is acting on enzyme-induced (or enzyme-facilitated) loops. Spontaneous loops can form anywhere on the long DNA molecule, possible a long distance from the bound topoisomerase. An enzyme-induced loop is formed where the topoisomerase is bound. The authors test the spontaneous loop hypothesis but none of the experiments rule out enzyme-induced (or enzyme-facilitated loops). Unless the authors can provide more convincing evidence for topoisomerase relaxing non-plectonemic DNA they should not make this claim. The biological implications for this unsubstantiated claim (expanded upon in the same paragraph) are meaningless.

8. There is additional confusion regarding "plectonemic" vs. "non-plectonemic" DNA. The sentence, "Taken together, our findings establish that topo II is much more stable on plectonemic DNA (high torsional stress) compared to on non-plectonemic DNA (low torsional stress)" makes it sound like "high torsional stress" cannot be "non-plectonemic," which is not true. High degrees of pure twist (high torsional stress) can be achieved with no writhe. The sentence also makes it sound as if high torsional stress must only be dealt with by writhe ("plectonemes"), which is not the case. For negative supercoiling, torsional stress is also readily relieved by formation of alternative DNA structures, kinking, base-flipping events, bubbling, and more.

9. The direction of this torsional stress for the "naked DNA" needs to be clearly stated up front. The use of "+ 35 turns and + 60 turns in figure 2 make it sound as if only positively supercoiling is done for the "naked DNA." But perhaps they meant "the addition of turns" (in the negative supercoiling direction?). This ambiguity must be made very clear. The precise degree of total torsional stress in these molecules must be stated, at least relatively (if it cannot be measured exactly, this must be acknowledged in the manuscript).

10. The term "hat curve" should either be explained clearly or, better, dropped.

11. The manuscript makes a bold claim about processivity two orders of magnitude greater than

previously determined. As the authors state, this degree of processivity could not be detected with previous experimental setups. This further illustrates the need to better explain the experimental setup and constant extension control.

12. The title for figure 1 is misleading: "Plectonemes stabilize topo II binding to DNA". The data in the figure does not measure binding.

13. The text: "suggesting that topo II has a much stronger affinity for chromatin over naked DNA" and Figure 3: "Chromatinization enhances topo II binding under low torsional stress" conflate binding and activity. The experiments measure activity. It is true that the enzyme must be bound to the DNA to be active, but the converse is not true. The enzyme can be bound but not active. The authors should be a lot more careful in their terminology.

14. The statement "Chromatin is the natural substrate of topo II in vivo" is too simplistic and potentially misleading. Estimates have been made on how much of the DNA is bound by proteins and although there is disagreement as to how much is not bound, all studies agree that some DNA is "free" unbound. Thus, a viable model is that yeast topo II acts on the unbound region only, which means that there is simply much less DNA available to it in the "chromatinized" experiments, but this fact is not mentioned.

15. In the text, "we again used the repeated winding experiment (Fig. 1b) but with chromatin fibers containing ~50 nucleosomes instead of naked DNA (Fig. 3b)" It is not immediately obvious from this statement or from the Methods which DNA substrate was used. Presumably the 64mer substrate containing tandem repeats of a nuclear positioning sequence was used. Firstly, the authors should explicitly mention in the main body of the text exactly which substrates were used for which experiments. Secondly, it appears that two very different DNA substrates were used for the naked DNA experiments and the chromatin experiments. The nuclear positioning sequence that is repeated in the 64mer DNA used for the chromatin experiments will affect the properties and behavior of DNA. The naked DNA experiments should be repeated using the 64mer substrate otherwise the comparison is not valid. Sequence and length effects are unlikely to explain all the differences but cannot be ruled out. The lengths of the DNA being used is not given in the text but should be. Later "~50 nucleosomes" are added to this DNA but still the length of DNA is not given in the text. One must dig through the Methods section and then the figure legends to try to figure out what was compared to what. It seems that the chromatinized DNA used was never tested "naked." But that should be the only DNA compared. Both length and sequence are changing in the comparison, yet this is not acknowledged. For the experiments with "chromatinized DNA", the comparator (non-chromatinized DNA) must be treated the same way (same DNA sequence and length and torsion put in both positive and negative directions).

16. In the Method section "active tethers" are mentioned, but it is not clear how the method that attaches the DNAs to the slides would be "active" or "inactive." They are tethers. Perhaps "active" is meant to indicate that it is bound to DNA? Or that it is the one being assessed by the researchers at the given time?

17. The assumption in the manuscript is that addition of nucleosomes increases DNA crossovers. But there are too many other variables changed to make that assumption. How do the authors rule out the possibility that nucleosomes interact with the yeast topo II (what biochemists refer to as "the BSA effect" because addition of BSA seems to enable/enhance every biochemical reaction)? Already the DNA length and sequence used for the "naked DNA" results used to generate the other rates to which the chromatinized DNA is compared to differ, but this length additionally changes when bound to ~50 nucleosomes but possible ramifications of this major change in the experimental set up is not mentioned.

18. Referring to the 12,688 bp double-stranded DNA (or later the 64 repeats of 601 NPE and 500 bp adaptors) as a "trunk" is unnecessary and oddly confusing.

19. How do the researchers know and how did they assess that two "flushes" of "5x chamber volume of buffer" is enough to wash away unbound enzyme? Is the efficiency of this wash assessed? If not, then the sentence "These two washes minimize further topo II binding events during measurements" should be changed to "might/are hoped to minimize further topo II binding events."

20. From the sentence, "If the number of topo II bound to a tether follows a Poisson distribution, this would correspond to ~90-95% of the active fraction of tethers being bound to a single topo II," is

there any published evidence that yeast topo II binds DNA with a Poisson distribution? If so, a reference(s) should be added. If not, is there any evidence of the opposite for the yeast or any topo II enzyme? Is there evidence that it aggregates or cooperatively binds DNA? If so, then a Poisson distribution is a bad assumption. That it is a bad assumption might be supported by the observation that 27% of the ~10–15% DNAs (not tethers as stated; yeast topo II is surely not interacting or binding to the tethers but to the DNA tethered to the slide by the tethers) is being relaxed 2–3 time faster. But that is assuming that each enzyme binding would be additive in activity. What if the 2–3 fold faster rates seen in nearly 1/3 of the experiments is the true rate and there are two states of topo II—a normal active state and a not as active state and only the not as active state was measured? I struggle to understand, given the experimental set-up, how two or three enzymes acting would yield a 2–3 fold faster rate. It would seem to me that a rate-limiting single enzyme activity would be the read out of the experiment. This section indicated a lack of critical evaluation by the researchers and, perhaps, a too rapid dismissal of a significant portion of the data.

21. Given that the length of DNA is known (although not clear to the readers in the manuscript) and the number of turns is known, why isn't there an attempt to quantify degree of supercoiling, or at least relative supercoiling (normalized for length)? It might not be absolute, but it would be highly quantitative relatively. Being more quantitative about degree of supercoiling, at least for the "naked" DNA experiments would be useful and allow quantitative comparisons to be made. If this cannot be done, it must be acknowledged. With the histone bound DNAs, quantification is more challenging, especially because it appears that histones allow "absorption of + supercoiling." In the absence of such quantitation, however, the authors cannot compare "chromatinized" to "non-chromatinized" DNA. There are too many other variables (different "trunks" used, different DNA sequence, different DNA lengths, shorter effective DNA length when chromatinized and thus less DNA available to enzyme, possible protein-topo II interactions, and more) for this comparison to be made. That all the experimental variables affecting the experiment were not acknowledged by the researchers is bothersome, again indicating a lack of critical thought.

22. The text states, "Taken together, our results suggest topo II dynamically senses torsional stress in DNA and adjusts its activity accordingly (Fig. 5)". But figure 5 does not indicate how topo may sense torsional stress. It shows how topoisomerase may recognize plectonemes. Plectoneme formation reduces torsional stress. It is the plectonemes that are being recognized (the consequence of torsional strain), not torsional stress itself.

23. There were a few sentences that contained editing/splicing errors problematic enough to make the meaning obscured. This is one example: "This released the "inner-turn" DNA (interacting primarily with the H3/H4 tetramer) was released from the histone core octamers, which is ~75 bp in length for each nucleosome."

24. In the text "based on source code available on Omar Saleh's website", who is Omar Saleh? The URL should be provided.

25. Equally odd were two comments in the editorial policy checklist: "We will include individual data points upon revision of the manuscript. Our current plots show the mean with an n in the range of 9 to 16." And "We will update our plots to show the data distribution upon revision. Our current plots show the mean with a clearly defined error bar." Why not just do it right in the first submission?

Overall, the study potentially provides some interesting observations. But in its current form it is let down by the issues described above.

Response to Referees

Re: NCOMMS-23-05291

“Chromatinization Modulates Topoisomerase II Processivity” by Lee et al.

We greatly appreciate the helpful comments and suggestions from the three referees and the Editor. We have carefully considered each comment and taken comprehensive actions to address them. These changes further strengthen the original conclusions and enhance the manuscript’s clarity.

Below, we begin with a summary of the major changes. We then provide a detailed, point-by-point response to each comment from the three referees, with each comment (bold) followed by our response (not bold). Note that in the revised manuscript, we have marked the suggested changes in red.

Summary of Major Changes:

Below, we summarize the most significant changes in the revised manuscript.

1. Referee #3 requested that we discuss our data in terms of torsion quantitatively. We think that this is an excellent idea. Our lab has a long history of making quantitative torsion measurements. Torsion, which measures the resistance to twist, should be characterized by torque. We have added a new Fig. 1 with data taken from our angular optical trap (AOT) to show direct torsional response of DNA to supercoiling. We have also added the relevant torque plots in the revised Fig. 4b (torque on pre-buckled DNA under different forces) and Fig. 5a (torque on a chromatin fiber).
2. Referee #2 and Referee #3 requested that we repeat processivity experiments on pre- and post-buckled naked DNA under (-) supercoiling. These new data have been added to Fig. 2 of the revised manuscript. The combined processivity data on (+) and (-) supercoiled DNA show that topo II is more processive on post-buckled DNA (higher torsional stress) than on the pre-buckled DNA (lower torsional stress), regardless of the torsion chirality.
3. Referees #2 and #3 pointed out that our processivity measurements of yeast topo II may not apply to other topo IIs. To investigate the generality of our findings, we have repeated the processivity experiments on post-buckled naked DNA using human topo II α and human topo II β . We have added a new main figure, Fig. 3 of the revised manuscript, to show that human topo II α and II β also show high processivity with the constant-extension method (>> 1000 turns), much longer than that previously reported for human topo II α by Seol et al. (JBC, 2013). Thus, we now have evidence that yeast topo II, human topo II α , and human topo II β all have long processivity, indicating that this may be a general property of the enzyme family.
4. Referee #3 requested direct evidence of topo II relaxation in the pre-buckled DNA. Inspired by this request, we have found a way to make a direct analysis and have included the results of this approach in Fig. 4b of the revised manuscript. This inclusion is exciting as it clearly demonstrates, without any ambiguity, how topo II relaxes pre-buckled DNA. This type of measurement has never been shown previously. We are excited about this inclusion.
5. In order to more clearly demonstrate the exceptionally long processivity of topo II on buckled chromatin fibers, we added a new main figure, Fig. 5 of the revised manuscript, to show individual

traces of topo II relaxation time course on (-) and (+) buckled chromatin fibers measured using the constant-extension method.

6. Referee #3 suggested that our processivity findings might depend on the DNA sequence and asked that we repeat some experiments using the naked 64mer DNA (the same DNA used for the chromatin studies in our work). We have conducted processivity experiments on the pre-buckled 64mer DNA under both (+) and (-) supercoiling. These new processivity data, now part of Fig. 6b and 6c, fully agree with those obtained with the initial DNA template containing no repeat sequences, suggesting the topo II behavior is insensitive to DNA sequence.
7. Referees #1 and #3 requested a more accurate method to evaluate the single topo II condition. We have found a more appropriate way to determine the probability of having a second topo on the same DNA molecule. We have discussed this on pages 27-28 of the revised Methods section.
8. Referee #2 commented on DNA writhe, which is a quantity related to, but not the same, as DNA crossing. We have added a new supplementary figure, Supplementary Fig. 5 of the revised manuscript, to show that a simple model considering DNA writhe (instead of DNA crossing) does not agree well with our measurements.
9. Referees #2 and #3 requested technical information on our constant-extension method. We have added a new supplementary figure, Supplementary Fig. 1 of the revised manuscript, to provide a schematic flow chart of the extension feedback control and algorithm. We also show representative raw traces of how the clamp responded to bead position changes, as well as validation of the method by the agreement of the topo II relaxation rate of post-buckled DNA measured with the constant-extension method and the conventional method.
10. Referee #2 and #3 suggested that a DNA bend introduced by a bound topo could increase the DNA crossing rate. We have repeated our simulations by considering such a bend. Indeed, the DNA crossing rate modestly increases after such consideration. We have shown these new simulation results in a new supplementary figure, Supplementary Fig. 7 of the revised manuscript, which is very similar to the simulation without a bend and can also explain our data well.

Referee #1 (Remarks to the Author):

In their manuscript, Lee and coauthors present exciting new single-molecule data on the activity of yeast topoisomerase II. They use cleverly designed and highly innovative new magnetic tweezers assays to assess topoisomerase activity on different substates (naked DNA, chromatin) with well-defined supercoiling / plectonemic states. Using this approach, some key and surprising new findings have been made: the processivity of topoll on plectonemic DNA is much larger than thought before. Furthermore, they provide the first quantitative assessment of topoll activity on "pre-buckled" DNA and show that topoll binds substantially shorter to this DNA and has a 2 orders of magnitude lower processivity. Using further experiments and Monte Carlo simulations they show that this difference can be understood in terms of DNA loops being required for topoll activity: DNA crossings or loops are always there in plectonemes and only intermittently, driven by thermal fluctuations in the pre-buckled state. Using a chromatin model, they show that chromatin, which has plenty DNA crossings, results in an increased topoll activity.

Taken together, this is a very nice manuscript: innovative new approaches, great, impactful and important findings. Furthermore it is very well written and accessible. I fully support publication in Nature Communications. Content-wise I do not have questions / remarks. The authors could consider looking into the two textual suggestions below, which might help to make the manuscript even more accessible.

We thank the Referee for these positive and encouraging comments.

1) 2nd paragraph of the results section. The authors conclude that "a single topoII molecule" remained active for 30 minutes. Here they do not explicitly explain why this conclusion can be drawn here. Later, for other experiment, they do discuss this. It might be helpful to also (or instead) discuss here why this statement can be made.

We thank the Referee for this helpful suggestion, which is related to a comment from Referee #3. Indeed, to enable the experiments described in this manuscript, we devoted significant attention to ensuring that our measurements mostly reflect the behaviors of a single topo II molecule by taking the following approaches:

- 1) We used a very low concentration of topo II (0.5 pM), while previous single-molecule studies typically worked with topo II concentrations in the nM to μ M range.
- 2) We flushed the sample chamber to remove free topo II in solution in the following way. After 2 min of topo II incubation, we flushed the sample chamber using a 5x chamber volume of the topo reaction buffer without topo II, waited for 5 min, and then repeated the flushing. These two washes reduce further topo II binding events during measurements. After this process, typically \sim 10-15% of our tethers showed topo II relaxation activity when wound down, which we defined as "active." Because of this low probability of a tether being initially active, the initial activity from these tethers was most likely from a single topo II molecule.
- 3) However, during the course of the subsequent measurement, an additional topo II molecule, which could be a consequence of any incomplete flushing, could become bound to these tethers and relax them. To determine the likelihood of this occurrence, we performed the conventional "wind down and wait" control experiment and monitored the tethers that were initially inactive but became active later in the measurements. We found this probability to be $\sim 17\pm 5\%$ over 30 minutes. Therefore, our measurements should be predominately from a single topo II.
- 4) For experiments using the constant-extension method, we were able to impose more stringent selection criteria to ensure a single topo enzyme condition. In this method, the activity of an additional topo II molecule caused an abrupt doubling in the relaxation rate. Thus, we could readily identify and exclude those traces from further analysis.

In the revised manuscript, we have expanded the discussion to include these points under Methods on pages 27-28 and provided a brief discussion in the main text on page 5.

2) "TopoII is less processivbe on ..." section. End of 2nd paragraph. I could not completely understand the sentences "The extension after each of those ... due to the short duration of the rewinding steps". Could the authors try to expand this discussion a little bit? The assay introduced here is super elegant,

but also somewhat complex. A. bit more explanation here, could help the general reader.

We agree with the Referee that our discussion of this method was unclear. We intended to convey that we estimated at most ~ 2.5 turns were relaxed by topo II during the brief 0.875 s rewinding step during which the DNA was buckled, far fewer than the 35 turns added (Fig. 4b of the revised manuscript).

In the repeated winding experiments to probe topo II activity on pre-buckled DNA (Figs. 2 and 4), the DNA was buckled immediately after each winding step. The measured extension right after the winding step was used to determine DNA supercoiling state in the preceding step (Supplementary Fig. 3). To do so, we determined the supercoiling state after the rewinding step, subtracted it from the number of turns added in the rewinding step, and accounted for relaxation of buckled DNA during the short rewinding step (typically ~ 2.5 turns). The resulting value provides the supercoiling state of the DNA in the preceding step.

Referee #2 (Remarks to the Author):

Lee and co-workers use magnetic tweezers to study the yeast topoisomerase II relaxation of positively supercoiled DNA in the pre- and post- buckled state along with the relaxation of positively and negatively supercoiled chromatinized DNA substrates. The authors observe a highly processive relaxation of positively supercoiled DNA in the post-buckled state while a lower processivity and lower binding time in the pre-buckled state. The relaxation in the pre-buckled state is attributed to the capture of spontaneous loops that the topoisomerase can capture. On chromatinized DNA the authors observe enhanced binding and relaxation processivity at low supercoiling and find that there are subtle differences in relaxation rate in three distinct phases of chromatin.

Overall these results provide some new insights into the activity of yeast type II topoisomerase – principally its high processivity, however the main results are largely incremental in nature as they modestly extend and largely confirm what is known about the mechanism of type II topoisomerases. This work is much better suited for a more specialized journal as laid out in detail below.

We thank the Referee for giving our manuscript a careful read.

- What are the noteworthy results?

This study slightly extends previous measurements of binding and relaxation activities of topo II in the prebuckling region of DNA supercoiling, particularly in the presence of chromatin (1) (2, 3), but largely confirms previous findings from many different methods that topo II preferentially binds to and relaxes DNA crossings (4-6).

This comment is related to another comment from this Referee. Please see our response below.

- Will the work be of significance to the field and related fields? How does it compare to the established literature? If the work is not original, please provide relevant references.

Although the work was not sufficient to fulfill the objective of this study to address the effect of DNA torsion on topo II activity, the work presents some very specific and detailed findings that could be of interest to the fields of DNA topology and topoisomerases.

We appreciate the Referee's recognition that our work could be of interest to the fields of DNA topology

and topoisomerases.

Specificially:

The authors claimed that there was no direct evidence for supercoil relaxation prior buckling by topo II.

“but how topo II activity responses to torsional stress is not fully understood. Previous biochemical and single-molecule studies have successfully elucidated many aspects of topo II activity on naked plectonemic DNA^{6, 15, 16, 17, 18, 19, 20}. However, torsion can also accumulate in DNA prior to plectoneme formation, and there has yet been no direct evidence for topo II relaxation of such a substrate. More importantly, even less is known about the processivity and rate of topo II relaxation of chromatin, the primary substrate of topo II in vivo.”

However, the subject has been addressed by Strick et al using *D. melanogaster* topo IIA who also provided the same conclusion that efficient topo IIA supercoil relaxation requires presence of DNA crossings or DNA writhe and that the relaxation rates would be limited prior to plectoneme formation (2) (p.606-607). Additionally, the same article showed the DNA crossover capture by topo II prior to buckling transition based on the elegant single molecule experiment and similarly for topo IV (7).

The publications noted by the Referee represent seminal contributions to the topoisomerase field. However, the literature does not provide clear evidence for topo II relaxation of pre-buckled DNA. Strick et al. (Nature, 2000) explicitly states, “our data clearly show that plectonemes are the enzymatic substrate. The enzyme is **unable** to relax supercoiled DNA efficiently below the buckling instability (data not shown).” Strick et al. (Comptes Rendus Physique, 2002) state, “the enzyme is **inefficient** at relaxing twist in a DNA molecule below the buckling transition.”

Furthermore, although Strick et al. (Nature, 2000) and Charvin et al. (Biophys J., 2005) found that topo II/topo IV trap loops in the pre-buckled DNA, they conducted experiments **without** ATP, and thus could not provide information on topo relaxation of pre-buckled DNA. Therefore, neither of these publications offers evidence of topo II relaxation of pre-buckled DNA.

Since the Referee stated, “efficient topo IIA supercoil relaxation requires the presence of DNA crossings or DNA writhe”, we want to point out that DNA writhe and DNA crossing are related quantities – a high writhe number typically corresponds to a high number of DNA crossings – but they are NOT proportional. In the pre-buckled regime, writhe increases linearly with turns as noted by the Referee (see Fig. 1c of the revised manuscript), while DNA crossings increase sharply with turns near the buckling transition (compare Supplementary Fig. 5 with Supplementary Fig. 6 of the revised manuscript).

To explain how topo relaxes pre-buckled DNA under different forces (Fig. 4b of the revised manuscript), we initially attempted to model the topo II relaxation rate dependence on writhe, by using writhe measured from our prior publication (Fig. 1c of the revised manuscript; also see Gao et al., Phys. Rev. Lett., 2021). In the pre-buckled regime, twist is proportional to writhe. However, we found that such a simple model considering writhe did not explain the observed relaxation behaviors of topo II. Given the comment from this Referee, we decided to show why this analysis does not work in the new Supplementary Fig. 5 of the revised manuscript. Instead, a model considering DNA crossings, which we obtained from MC simulations, shows good agreement with our data (Fig. 4b of the revised manuscript). We have also briefly discussed this under the main text on page 13 of the revised manuscript.

Thus, with the combination of extensive experimental data and rigorous theoretical modeling, our work significantly advances the current understanding of topo II behavior on pre-buckled DNA. This knowledge is important, as it shows for the first time how the enzyme is capable of quickly responding to low level's changes in supercoiling that might arise from housekeeping functions, such as basal levels of transcription or the action of looping enzymes such as SMC proteins.

Investigating the relaxation rate and the binding stability of topo II prior to a buckling transition might be good to understand the effect of chromatin on DNA topological state and geometry and the subsequent topo II activity. However, it is not sufficient to understand torsional effects on topo II supercoil relaxation activity. For topologically constrained DNA, DNA torsion increases linearly as increasing twist (T_w) in terms of the bead rotation prior to plectoneme formation (buckling transition) and this stored torsion will remain roughly constant after the buckling transition. Thus, it seems inadequate to study the effect of torsion on topo II activity prior to buckling transition as torsion would vary dependent on the residual twist while topo II removes supercoils. To address the question posed regarding the effect of DNA torsion on the activity of type II topoisomerase would require a systematic study of the rate and processivity of relaxation under differing levels of well-controlled constant torsion, which has been done previously for other type II topoisomerases. The current work explores only a single value of positive torsion in the supercoiling regime and poorly defined positive torsion in the pre-buckling regime.

In order to understand how topo II reacts to a different level of torsion of supercoils, I would recommend the authors to perform under a range of different torques for both positive and negative supercoils as done for other topoisomerase studies (3, 8) since supercoiling at different tension with magnetic tweezers leads to a different torsion for plectoneme DNA. Only torsion not tension would apply for topo II activity if topo II binds to DNA plectoneme during relaxation.

We very much appreciate the discussion of twist-writhe partitioning from the Referee. Indeed, added turns to DNA are partitioned to twist and writhe. As the Referee noted, DNA torsional stress is related to the twist in DNA and increases linearly with turns before buckling. In fact, this partitioning was of sufficient importance that it motivated us to develop a new method of direct torque measurements and theoretically model DNA torsional elasticity. Our publication (Gao et al., Phys. Rev. Lett., 2021) shows how turns partition between twist and writhe under different forces. To our knowledge, this publication represents a state-of-the-art understanding of this subject.

We also appreciate the Referee's request to explore topo II activities under different levels of torsional stress. DNA torsional stress is best characterized by torque (directly proportional to twist), which measures the resistance to twisting. Our lab previously conducted extensive direct torque measurements on a diverse range of DNA substrates, allowing us to conduct and report torque measurements in a new Fig. 1 (naked DNA), a revised Fig. 4b (naked pre-buckled DNA under different forces), and a revised Fig. 5a (chromatin). Including torque plots directly shows that our studies were conducted under different levels of torsional stress, and we thank the Referee for this request.

The Referee also asked to see experiments under different forces, which was precisely what was shown in Fig. 2b of the initial manuscript (and retained in Fig. 4b of the revised manuscript). We determined the relaxation rate of topo II on post-buckled DNA at three different forces and indicated them as the blue dashed lines. Fig. 4b of the revised manuscript also shows that supercoil relaxation by topo II slowed down when relaxing supercoils in the pre-buckled regime, as torsion (torque) decreased. Interestingly, these results further demonstrate that the topo II relaxation rate is not solely determined

by torsion. Instead, the observed topo II relaxation rate is consistent with a model that factors in DNA crossings.

To expand our studies in the torsional stress parameter space, we followed the request from the Referee to investigate topo II behavior under (-) supercoiling, repeating processivity experiments on pre- and post-buckled naked DNA under (-) supercoiling. As shown in Fig. 2 of the revised manuscript, we found that yeast topo II shows similar activity and processivity on (+) and (-) supercoiled DNA, and that this finding holds for pre- and post-buckled DNA.

All these experiments demonstrate the generality of our conclusions. Further expansion to include all possible conditions is beyond the scope of the current work.

If the authors are keen on understanding how crossing probability affects topo II enzyme rate, I would recommend the authors to employ the single molecule braiding technique that was pioneered by the Bensimon-Croquette group (9).

We appreciate this suggestion. Topo II relaxation of braided DNA would be a very interesting topic to explore in the future; however, it is beyond the scope of the current effort, which focuses on supercoiling relaxation within single DNA substrates by topo II.

- Are there any flaws in the data analysis, interpretation and conclusions? - Do these prohibit publication or require revision?

Binding time of topo II for prebuckled region estimation is acceptable despite probing with periodic magnet turns, but the processivity estimate obtained from this approach does not reflect the relaxation processivity of prebuckled DNA by topo II as the measured processivity was from the relaxation of 30 turn that encompass supercoils for both pre and post buckling transition. Furthermore, it wasn't the enzyme that stopped relaxation during the waiting time between turns. Put simply the approach can measure the binding time in the pre-buckled state but cannot accurately provide a measure of the processivity in this state.

We understand this concern from the Referee. We realized that we did not clearly explain how we determined the processivity of the pre-buckled regime in the initial manuscript. We determined the number of turns relaxed in the pre-buckled state during each repeat cycle by removing any turns relaxed in the post-buckled regime from the total 30 turns applied and considering only the turns relaxed in the pre-buckled DNA after; hence, we indeed only consider the turns relaxed in the pre-buckled regime. We have clarified these points in the main text on page 8, under Methods on page 29, and in Supplementary Fig. 2, of the revised manuscript.

Please note that during this revision, we have provided direct evidence for topo II relaxation of pre-buckled DNA in Fig. 4b of the revised manuscript. We show that topo II processively relaxed pre-buckled DNA while slowing down as it approached the zero-turn state.

I strongly recommend the authors should perform binding time measurements on negatively supercoiled DNA prior and post buckling transitions in order to compare the binding time of negatively supercoiled chromatin DNA data.

We thank the Referee for the suggestion. As requested and mentioned above, we have performed new

experiments to determine the processivity and active lifetime on pre- and post-buckled (-) supercoiled DNA. We found a similar activity lifetime and processivity on (-) and (+) supercoiled DNA, suggesting that these properties are primarily sensitive to the availability of DNA crossings and not the chirality of supercoiling. We have added these new data to Fig. 2 and discussed them in the main text on page 10 of the revised manuscript.

- Is the methodology sound? Does the work meet the expected standards in your field?

The constant extension technique could be an ideal way to maintain the level of supercoiling substrate to be constant for topoisomerase if the spatial and temporal resolutions are higher than the enzymatic activity. If not, the error of enzymatic rate estimation that was obtained from the turn data in the constant-extension would be couple to the extension measurement error per each turn due to the limited spatial and temporal resolutions. I would recommend the authors to validate the constant extension technique by comparing the rates from the constant torque method where a constant number of turns will be applied when the extension reaches 90% relaxation of plectoneme.

We thank the Referee for these suggestions. To address them, we have characterized the feedback performance by characterizing the feedback response and performing control experiments. We have now included these data as the new Supplementary Fig. 1.

These data show that the feedback had a response time of 0.23 s to a change in the DNA extension corresponding to 2 turns. To further validate the feedback method, we performed an experiment mimicking topo supercoiling relaxation at 3.0 turns/s and recovered the correct rate. We also verified that the topo II rate on buckled DNA measured using this method agrees well with that obtained using a standard method without any feedback.

- Is there enough detail provided in the methods for the work to be reproduced?

The descriptions of the methods were adequate.

We are glad the Referee found the descriptions of the methods adequate.

Detailed comments and concerns

Page 4.

“In the present work, we developed novel single-molecule assays that enable the direct..

... These experiments demonstrate the impact of varying degrees of torsion on topo II activity on plectonemic, non-plectonemic, and chromatinized DNA.”

Firsthand, it seems inappropriate to consider the employed single-molecule assay to be “novel” as many similar magnetic tweezers based single molecule assays have been performed. This is a small extension over existing feedback approaches that maintain the DNA under a constant level of torsion in the plectoneme regime. Also I do not agree that the work showed varying degrees of torsion on topo II activity, as described in detail above.

We have followed the suggestion of the Referee and removed all mentions of “novel” from the manuscript.

However, we are perplexed why the Referee does not think our work shows how varying torsion directly impacts topo II activity. The manuscript provides a plethora of data to support this conclusion. As just one example, Fig. 4b of the revised manuscript shows that topo II slows down as it continues to relax DNA in the pre-buckled regime against reduced torsional stress (note the new torque axes to the right of

each plot). We hope the added torque plots to Figs. 1, 4b, and 5a help highlight the clear torsional dependence of topo II activity.

Page 5.

“Unexpectedly, we discovered that topo II was much more processive on plectonemic DNA than previously thought..”

This claim requires a reference instead of indicating “previously thought”. This claim is made repeatedly throughout the manuscript but is never substantiated by a citation.

We agree that this sentence requires a reference. We have now cited Seol et al. (JBC, 2013), which was performed on human topo II α , on page 6 of the revised manuscript.

Page 8.

The authors make the following disingenuous claim in which the processivity of yeast topo II measured in the current work is compared to the processivity of other type II topoisomerases as measured in the cited work.

“This lower bound on the processivity far exceeds values suggested by other single-molecule methods^{15, 19, 20}, indicating that topo II can be more processive on plectonemic DNA than previously thought.”

The difference in processivity is a difference among enzymes rather than a difference among measurements, but the authors are confusing these two factors in a way that could be misleading. They would do better to state that the yeast enzyme is more processive than other type II topoisomerases, which is a valid statement that is one of the clear results of this work.

We understand the Referee’s concern about generalizing our yeast topo II processivity. We found yeast topo II to be much more processive than other type IIA topoisomerases examined in previous studies and wondered if this difference was due to a difference in topoisomerases or methods. To differentiate between these two possibilities, we conducted new experiments to determine the processivity of human topo II α and human topo II β on buckled DNA. These new data show that topo II α and topo II β also have processivity \gg 1000 turns, which is much greater than that previously measured for human topo II α (Seol et al., JBC, 2013). These data are now included in the new Fig. 3. Thus, our work establishes that yeast topo II, human topo II α , and human topo II β all have high processivity using our method. We have discussed this finding on pages 6 and 9 of the revised manuscript.

Page 9.

The authors state: “Taken together, our findings establish that topo II is much more stable on plectonemic DNA (high torsional stress) compared to on non-plectonemic DNA (low torsional stress).” This is incorrect. The torsional stress in the DNA is related to the twist whereas the formation of plectonemes is related to writhe in the DNA. In general this statement is completely incorrect and misleading. Under the same applied force, the torsion in the non-plectoneme regime is less than or equal to the torsion in the plectoneme regime, but without specifying this condition the authors have made an incorrect and misleading statement.

We initially intended to use the quoted sentence to contrast topo II behavior in pre and post-buckled regimes. As shown in the new Fig. 1 of the revised manuscript, under a constant force, DNA is under lower torsional stress in the pre-buckled regime, with torsional stress increasing linearly until the onset

of the buckling transition. Upon buckling, DNA is under higher torsional stress, which does not increase with added turns. Upon reading the comment from this Referee, we realized the quoted sentence might have a second interpretation, different from our intention. That is, a buckled DNA molecule contains both plectonemic DNA and non-plectonemic DNA, all under the same torsional stress. We thank the Referee for bringing this interpretation to our attention. We have removed most of the mentions of plectonemes in the revised manuscript to avoid this confusion.

Page 10.

“used to precisely calculate the turn state from the hat curve.. precisely probe..” I would recommend to remove “precisely” and also explain how precisely the authors calculated the turn state from the hat curve.

We have removed the word “precisely” from both instances. We have also further clarified how we calculate the supercoiling state from the extension-supercoiling state relation (hat curve) in the buckled regime on page 29 in the revised Methods section.

Page 16.

the authors make the following statement: “We unexpectedly found that topo II relaxed plectonemic DNA with much greater stability and processivity than previously thought (Fig. 1). These values were not measurable with previous single-molecule techniques.” This is inaccurate as similar feedback approaches have been used to measure highly processive relaxation event by type II topoisomerases (10).

The Referee stated that there have been similar feedback approaches. We very much appreciate the work by Seol et al. (JBC, 2013) which presents an excellent effort towards controlling the topological state of the DNA. However, they did *not* use any feedback control. Instead, they allowed topo to relax within the buckled regime until reaching the buckling transition. They then reset the turns to bring the extension back into the buckled state by adding a fixed number of turns. Thus, the topological state of the DNA varied during their measurements. In contrast, we maintain a constant post-buckled state by maintaining a constant DNA extension via a feedback algorithm.

Furthermore, Seol et al. (JBC, 2013) measured the processivity of human topo II α to be 168 turns using their method. In contrast, as mentioned above, our data taken with our feedback method show that human topo II α has a processivity \gg 1000 turns (the new Fig. 3). We feel that our method allows for more reliable processivity measurements on buckled DNA by providing topo II with a more consistent DNA topology.

Page 25.

I would recommend the authors to show some raw traces for implementing the constant-extension clamp.

We thank the Referee for this suggestion. We have now provided raw traces to characterize the performance of the constant-extension feedback clamp in the new Supplementary Fig. 1.

Page 28 and 37 (figure 2C).

It is unclear based on the method what the two fitting parameters are. Please indicate in the main text what they are and their error bars. Furthermore – the MC simulations ignore the effect of the

type II topoisomerase bound to the DNA in inducing the bending and buckling of the DNA that is well established in the literature (7). Given that the type II topoisomerase is thought to remain bound in the pre buckled state, it will contribute significant bending energy leading to increased rate and probability of buckling in this state. This fundamental contribution of the binding of the type II topo to the DNA in driving the spontaneous looping was completely ignored in the simulations and in the related discussion.

In order not to exceed word limit for a figure caption, we decided to discuss the fitting parameters and provide the fit values and their error bars under the Methods section of the revised manuscript (page 33).

As the Referee noted, our simulations do not consider topo II-induced DNA bending. To evaluate the contribution of such a bend to the rate of crossover formation, we performed new simulations by introducing a bend in the middle of the DNA with a bend angle of 120 degrees, comparable to that introduced by yeast topo II (Dong and Berger, Nature, 2007; Hardin et al., NAR, 2011; Lee et al., NAR, 2013). The new results are shown in a new Supplementary Fig. 7. As shown in this figure, the introduction of a bend indeed increases the looping rate as predicted by the Referee. The resulting simulations also show good agreement with measurements.

Figure 1D. As stated previously, the measurement of the pre-buckled relaxation activity can not provide a measure of the processivity of the enzyme in this regime.

As explained above, the data from pre-buckled relaxation actually can measure the enzyme's processivity in this regime. Please see this explanation.

Figure 3. There is no caption for part D.

We thank the Referee for catching this glaring omission! We have added a caption for the original Fig. 3d, which is now Fig. 6c of the revised manuscript.

Figure S6. Adjust the x and y axis ranges to >40 turns in order to make data more visible.

We thank the Referee for this suggestion. In the initial manuscript, the x- and y- axis ranges started from zero to show that all arrays contained about 50 nucleosomes. This initial plot was too small to make the relevant region visible. We now also show a zoom-in of the region > 40 nucleosomes. This SI figure is now Supplementary Fig. 9 of the revised manuscript.

In summary, the authors have measured the rate and processivity of yeast topo II on positively supercoiled DNA in the pre-buckled and plectoneme states. They also measure the rate and relaxation of chromatized DNA under both negative and two distinct phases of positive supercoiling conditions. Their conclusions summarized in figure 5 are that yeast topo II remains bound longer, and is more active, on DNA molecules containing crossings. These results largely confirm what is known about type II topoisomerases, but extend the field to provide detailed rates and processivity measurements for yeast topo II, which has not been extensively studied with similar techniques. Although these measurements of yeast topo II activity will be of utility and interest to the community, the results do not provide significant broader insights or mechanistic details concerning type II activity to warrant publication in Nature Communications.

Our response to this Referee above should have clarified the novelty aspects of the work: a new method to measure topo activities under a constant topological state, exceptionally long processivity of yeast topo II (and also of human topo II α and human topo II β), direct observation of topo II relaxation of pre-buckled DNA, and direct measurements of topo processivity and rate on chromatin. These findings expand our understanding of topoisomerases and provide a more comprehensive and dynamic picture of topoisomerase action. They set a new bar for assessing how torque affects DNA motor proteins and hence will be of interest to those working on enzymes pertinent to replication and transcription as well.

Literature cited:

1. Le TT, Gao X, Park Sh, Lee J, Inman JT, Lee JH, Killian JL, Badman RP, Berger JM, Wang MD. Synergistic Coordination of Chromatin Torsional Mechanics and Topoisomerase Activity. *Cell*. 2019;179(3):619-31.e15. doi: <https://doi.org/10.1016/j.cell.2019.09.034>.
2. Strick TR, Charvin G, Dekker NH, Allemand J-F, Bensimon D, Croquette V. Tracking enzymatic steps of DNA topoisomerases using single-molecule micromanipulation. *Comptes Rendus Physique*. 2002;3(5):595-618. doi: [https://doi.org/10.1016/S1631-0705\(02\)01347-6](https://doi.org/10.1016/S1631-0705(02)01347-6).
3. Strick TR, Croquette V, Bensimon D. Single-molecule analysis of DNA uncoiling by a type II topoisomerase. *Nature*. 2000;404(6780):901-4.
4. Roca J, Berger JM, Wang JC. On the simultaneous binding of eukaryotic DNA topoisomerase II to a pair of double-stranded DNA helices. *J Biol Chem*. 1993;268(19):14250-5. doi: [https://doi.org/10.1016/S0021-9258\(19\)85234-1](https://doi.org/10.1016/S0021-9258(19)85234-1).
5. Roca J, Wang JC. The probabilities of supercoil removal and decatenation by yeast DNA topoisomerase II. *Genes to Cells*. 1996;1(1):17-27. doi: 10.1046/j.1365-2443.1996.01001.x.
6. Zechiedrich EL, Osheroff N. Eukaryotic topoisomerases recognize nucleic acid topology by preferentially interacting with DNA crossovers. *The EMBO Journal*. 1990;9(13):4555-62. doi: <https://doi.org/10.1002/j.1460-2075.1990.tb07908.x>.
7. Charvin G, Strick TR, Bensimon D, Croquette V. Topoisomerase IV bends and overtwists DNA upon binding. *Biophys J*. 2005;89(1):384-92. Epub 2005/05/03. doi: 10.1529/biophysj.105.060202. PubMed PMID: 15863484; PMCID: PMC1366538.
8. McKie SJ, Desai PR, Seol Y, Allen AMB, Maxwell A, Neuman KC. Topoisomerase VI is a chirally-selective, preferential DNA decatenase. *eLife*. 2022;11:e67021. doi: 10.7554/eLife.67021.
9. Neuman KC, Charvin G, Bensimon D, Croquette V. Mechanisms of chiral discrimination by topoisomerase IV. *Proc Natl Acad Sci USA*. 2009;106(17):6986-91. doi: 10.1073/pnas.0900574106.
10. Seol Y, Gentry AC, Osheroff N, Neuman KC. Chiral discrimination and writhe-dependent relaxation mechanism of human topoisomerase II α . *J Biol Chem*. 2013 May 10;288(19):13695-703. doi: 10.1074/jbc.M112.444745. Epub 2013 Mar 18. PMID: 23508957; PMCID: PMC3650406.

Referee #3 (Remarks to the Author):

Lee et al. *Nature Communications* 2023/03

This is an interesting manuscript where a new single molecule technique was devised to enable continuous monitoring of yeast topo II relaxation to test how the enzyme responds to differing degrees of torsional stress. The new technique allowed the researchers to hold the winding of DNA constant (by spinning it as the enzyme relaxes it). This is as opposed to previous work where the enzyme relaxes plasmid (with a limited number of supercoils) and slows down as the plasmids become less supercoiled, thus giving different (and unmeasurable) rates over the different supercoiling ranges. The researchers go on to assess what seems to be a different DNA with chromatin. There are interesting results in the manuscript, but some conclusions are not supported by the evidence, previous work relevant to the new results is ignored, the manuscript is written for experts in single molecule experiments, experimental procedures are not well explained, and the authors use imprecise terminology.

We thank the Referee for the insightful comments and valuable perspectives on our work. These helpful suggestions have strengthened our conclusions and improved the clarity of our work.

Specifically:

1. Many previous publications revealed a strong dependence of topo II (from various organisms) for supercoiled DNA over relaxed DNA. And some have demonstrated a strong preference for one supercoiling handedness (+ versus -) over another. Yet all this previous work is ignored. The researchers must fit their new results into the known work critically.

We thank the Referee for this suggestion. We have added several references on the preference of topo II relaxation of supercoiled DNA over relaxed DNA on pages 3 and 10 of the revised manuscript. These references are helpful as we discuss our results in the context of existing work.

During the revision process, we have also conducted new topo activity and processivity experiments on (-) supercoiled DNA, and these new data are added to Fig. 2 of the revised manuscript. However, our data do not show any strong chirality dependence of yeast topo II. This finding is consistent with a prior biochemical study that we now cite on page 10 of the revised manuscript.

2. The question of the generality of the observations is important and is too superficially dealt with by the authors in the manuscript discussion, “As all type IIA topoisomerases share the same core mechanism, the properties of yeast topo II revealed by methods used in this study may be consistent in all type IIA topoisomerases, including those in higher-order organisms.” This question is so easily addressed by assessing different (preferably “higher order”) topo II. Importantly, many studies (ignored in the manuscript) have comprehensively compared activities, including relaxation, by various topo II’s and the differences across species are striking. To which enzyme, then, the researchers compare their new rates is not clear.

We took this comment, and a related comment from Referee #2, to heart. Although it is beyond the scope of the current work to comprehensively compare all properties of different topo IIs, we decided to focus on one property that our new constant-extension method was uniquely poised to measure: the processivity on buckled DNA. Do human topo IIs also have high processivity on buckled DNA? We carried out new experiments and found that human topo II α and topo II β have processivity \gg 1000 turns, which is much greater than that previously measured for human topo II α by Seol et al. (JBC, 2013). These data are now included in the new Fig. 3. Thus, our work shows that yeast topo II, human topo II α , and human topo II β have much higher processivity than previously reported for human topo II α (Seol et al., JBC, 2013). We have also discussed these findings on pages 6 and 9 of the revised manuscript.

3. In the abstract, what is novel about the single molecule set-up should be mentioned. The experimental setup, which is novel and crucial to the study, should be explained clearly in the text. The diagram of the setup in Figure 1a is not helpful as it looks like any other magnetic tweezers experiment. How is the constant extension maintained? A schematic of the extension correction protocol (as part of figure 1) is needed. The description of the constant extension clamp is impossible to follow in the Methods.

We also wanted to clarify the novelty of the constant-extension method in the abstract. Unfortunately, we could not find a suitable way to do this, as we are constrained by the word limit of the abstract.

We also agree that providing more technical details of the constant-extension method is helpful, as this is the first report of such a method. We have included this information by creating the new Supplementary Fig. 1 of the revised manuscript. As suggested by this Referee, we have included a

schematic flow chart of the extension feedback flow and algorithm. In addition, this figure also shows representative raw traces of how the clamp responds to bead position changes. We have further validated the method by demonstrating that the topo II rate measured with this method is consistent with that obtained using a conventional single-molecule topoisomerase method.

4. The abstract would be improved by more precision in the measures. How much “faster”? The data generated are highly quantitative, so the abstract, indeed the entire manuscript, should be as well.

We thank the Referee for this suggestion. We have modified the abstract to be more quantitative.

5. The authors use a term “pre-buckling” (and “pre-buckled”) (to contrast with “plectonemic”). But the comparisons are not clear. Further, well defined terms apply to the situation. It seems that the authors are describing supercoiling that is only twist (pre-buckling) versus supercoiling that is only writhe (buckling? = plectoneme?). I would not introduce or use these less precise terms to describe what there are already terms for. If I am wrong (that “pre-buckling” = twist and “buckling” = plectoneme = writhe), then a much better explanation is needed to help readers follow. It is especially confusing terminology because “pre-buckled” could mean that the buckle was put in before the experiment (this is how I read it the first time through the manuscript). Or it could mean that it has not yet “buckled,” which must be what is meant. But few readers will work as hard as I did to understand and follow. The loose term becomes even more problematic when used in the context of nucleosome binding. The sentence “Each nucleosome contributes about +1.0 turns to the (+) width of the hat tilt, implying that each nucleosome can absorb about 1.0 turns of (+) supercoiling before buckling” is read that the nucleosome buckles. Is that what was meant? Later, in the Methods section, a new term “buckling-like” is introduced. Because “buckling” is already a fuzzy term, what is meant by this term is even fuzzier.

We sincerely thank the Referee for their extraordinary efforts in understanding our work. We agree with the Referee that the terminologies can be confusing. Although we use the conventional terminology in the field of DNA topology, we understand that this may not be common knowledge. Upon reading the comment from the Referee, we realize that we should explain these terms within the manuscript. Therefore, we have added a new main figure and made it Fig. 1 of the revised manuscript. This figure shows the direct torque measurements using our angular optical trap. These data allow us to show how “buckling” is defined, what are the pre- and post-buckling states of DNA, and how supercoiling partitions into twist and writhe.

As shown in this figure, when DNA is held under a low constant force while turns are added to DNA, there are two primary regimes of DNA behavior. Prior to plectoneme formation (i.e., in the pre-buckled regime), turns are partitioned between twist and writhe. The fraction of added turns going into twist or writhe is constant. Thus, twist and writhe increase linearly with added turns. As more turns are added to DNA, DNA buckles by extruding a plectoneme – this is the definition of “buckling”. The onset of the buckling transition occurs when the extension begins to linearly decrease with added turns while torque plateaus. Any additional turns added will be converted solely to writhe after plectoneme formation (i.e., in the post-buckled regime) and twist will remain constant with additional turns, whereas writhe will increase linearly as turns are added. Therefore, a plectonemic DNA contains both twist and writhe.

To remove torsion from DNA, topo II must act on DNA crossing. DNA crossings are related to writhe, but the two are not the same. In the pre-buckled regime, writhe increases linearly with turns, whereas DNA crossings do not. DNA crossings increase sharply near the buckling transition (compare Supplementary

Figs. 5a and 6).

The Referee also raised an excellent point related to chromatin topology. Unlike naked DNA, the topological properties of chromatin are poorly understood theoretically. Based on our experimental work (Le et al., Cell, 2019; also this work), the extension-turn relation of a chromatin fiber under a constant force also shows a rather flat extension region followed by a linear decrease in extension. This transition is reminiscent of the buckling transition of naked DNA, and we thus refer to it as “buckling-like”. This buckling-like transition possibly corresponds to the formation of a plectonemic structure formed by the entire chromatin fiber. We found that the transition occurs at ~ 50 turns for a chromatin fiber containing 50 nucleosomes (Supplementary Fig. 8), suggesting that each nucleosome absorbs about 1 turn before buckling. We have now clarified this on page 14 of the revised manuscript.

6. The problem is that in these studies, the only read out is the length of the DNA and the rapidity of the rotor. Forthright acknowledgement of the limitations of the experimental set-up would be helpful for the non-experts.

As the Referee noted, using only the extension and rotation as readouts may present some limits to this method. We have now noted that combining our approach with fluorescence methods could expand the utilities of this approach on page 22 of the revised manuscript.

7. The argument that topo II can relax “pre-buckled” DNA is not supported by data. Unless the authors provide more convincing evidence for this assertion, this claim should be dropped. To detect relaxation the researchers needed to rotate the magnets “to bring the molecules slightly into the plectonemic regime”. The most likely explanation is that the topoisomerase is acting on the plectonemes formed during the “brief activity check”. Not surprisingly the average processivity (~ 30 turns) corresponds to the number of turns added during the “brief activity check”. The authors imply that the topoisomerase is acting on the non-plectonemic region of the DNA molecule. More likely explanations are that either the topoisomerase briefly dissociates and rebinds to a crossover, or the plectoneme is formed at the place on the DNA where the topoisomerase is bound. Yeast topoisomerase II imparts a significant bend in the DNA. This bending likely facilitates the formation of the plectoneme. If the newly introduced plectoneme is formed at the site where topoisomerase is bound, the enzyme should act very efficiently. The authors speculate that topo II is capturing spontaneous loops that form due to thermal fluctuation. While this explanation is feasible, it seems more likely that the enzyme is acting on enzyme-induced (or enzyme-facilitated) loops. Spontaneous loops can form anywhere on the long DNA molecule, possible a long distance from the bound topoisomerase. An enzyme-induced loop is formed where the topoisomerase is bound. The authors test the spontaneous loop hypothesis but none of the experiments rule out enzyme-induced (or enzyme-facilitated loops). Unless the authors can provide more convincing evidence for topoisomerase relaxing non-plectonemic DNA they should not make this claim. The biological implications for this unsubstantiated claim (expanded upon in the same paragraph) are meaningless.

We very much appreciate this critical comment from the Referee. We agree with the Referee that not having direct evidence for the ability of topo II to relax pre-buckled DNA was unsatisfying. The Referee’s comment inspired us to think more rigorously about our data. It occurred to us that the traces of topo II relaxation into the pre-buckled regime (presented in Fig. 4a of the revised manuscript) contain all necessary information for direct determination of the turn-state in the pre-buckled regime using the DNA extension-supercoiling state relation. This is a straightforward method that we regret not using initially. We now show the result of this analysis in Fig. 4b and discuss this method under Methods on

page 30 of the revised manuscript.

As shown in the revised Fig. 4b, the direct and indirect methods are in good agreement. This agreement is satisfying and validates both methods! We also want to clarify the indirect method. For example, for the 15 s wait time, the "activity check" is brief (0.875 s), with topo II relaxing at most ~2.5 turns out of the 35 turns added during rewinding. To eliminate the resulting bias introduced during the brief activity check, we removed the ~2.5 turns when calculating the pre-buckled state. We have clarified this in the main text on page 11 and under Methods on page 29 of the revised manuscript.

We do not fully understand how the bending of DNA by topo II facilitates supercoil relaxation. To evaluate the contribution of this bend to the rate of DNA crossing formation, we performed new simulations by introducing a bend with a bend angle of 120 degrees in the middle of the DNA, comparable to that introduced by yeast topo II (Dong and Berger, *Nature*, 2007; Hardin et al., *NAR*, 2011; Lee, et al., *NAR*, 2013). The new results are shown in the Supplementary Fig. 7. These simulations show that a DNA bend indeed increases the crossing formation rate as predicted by the Referee. They also show good agreement with our experimental data. Although this is a first step towards understanding the relationship between enzyme-mediated DNA bending and crossing formation, a full understanding will require many new experiments and modeling work, which are beyond the scope of the current work and will be a goal of our future research.

8. There is additional confusion regarding "plectonemic" vs. "non-plectonemic" DNA. The sentence, "Taken together, our findings establish that topo II is much more stable on plectonemic DNA (high torsional stress) compared to on non-plectonemic DNA (low torsional stress)" makes it sound like "high torsional stress" cannot be "non-plectonemic," which is not true. High degrees of pure twist (high torsional stress) can be achieved with no writhe. The sentence also makes it sound as if high torsional stress must only be dealt with by writhe ("plectonemes"), which is not the case. For negative supercoiling, torsional stress is also readily relieved by formation of alternative DNA structures, kinking, base-flipping events, bubbling, and more.

We appreciate this comment from the Referee. Upon reading the comment from this Referee, we realized the quoted sentence might have a second interpretation, different from our intention. That is, a buckled DNA molecule contains both plectonemic DNA and non-plectonemic DNA, all under the same torsional stress. We thank the Referee for bringing this interpretation to our attention. We have removed most of the mentions of plectonemes in the revised manuscript to avoid this confusion and instead use the term "buckled DNA."

9. The direction of this torsional stress for the "naked DNA" needs to be clearly stated up front. The use of "+ 35 turns and + 60 turns in figure 2 make it sound as if only positively supercoiling is done for the "naked DNA." But perhaps they meant "the addition of turns" (in the negative supercoiling direction?). This ambiguity must be made very clear. The precise degree of total torsional stress in these molecules must be stated, at least relatively (if it cannot be measured exactly, this must be acknowledged in the manuscript).

"+ 35 turns and + 60 turns" are turns added and not the supercoiling state of the DNA. We indeed only applied (+) winding on naked DNA for Fig. 4a of the revised manuscript. With the addition of Fig. 1, we hope the direction of turns is more clearly defined.

The Referee asked us to clarify torsional stress, which is related to Comment 5 from this referee. We

have acted on the suggestion and added a y -axis to the right of each plot in Fig. 4b to show torque (which corresponds to the torsion in DNA) using data taken under the same buffer condition from our previous publication (Gao et al., PRL, 2021). For consistency, we have also now included torque data in Fig. 5a for a chromatin fiber using data taken under the same nucleosome saturation level and buffer condition from our previous publication (Le et al., Cell, 2019).

10. The term “hat curve” should either be explained clearly or, better, dropped.

Although the term “hat curve” is conventional in single-molecule topology studies, we understand it is a rather specialized term. We have removed its use from the manuscript and replaced it with “extension-supercoiling state relation”.

11. The manuscript makes a bold claim about processivity two orders of magnitude greater than previously determined. As the authors state, this degree of processivity could not be detected with previous experimental setups. This further illustrates the need to better explain the experimental setup and constant extension control.

This comment is related to comment #3 from the same Referee. We hope that our response to comment #3 satisfactorily addresses this comment as well.

12. The title for figure 1 is misleading: “Plectonemes stabilize topo II binding to DNA”. The data in the figure does not measure binding.

We thank the Referee for pointing this out. We have changed “binding to” to “activity on” to correct this.

13. The text: “suggesting that topo II has a much stronger affinity for chromatin over naked DNA” and Figure 3: “Chromatinization enhances topo II binding under low torsional stress” conflate binding and activity. The experiments measure activity. It is true that the enzyme must be bound to the DNA to be active, but the converse is not true. The enzyme can be bound but not active. The authors should be a lot more careful in their terminology.

As with comment 12, we have changed “binding” to “activity” to correct this.

14. The statement “Chromatin is the natural substrate of topo II in vivo” is too simplistic and potentially misleading. Estimates have been made on how much of the DNA is bound by proteins and although there is disagreement as to how much is not bound, all studies agree that some DNA is “free” unbound. Thus, a viable model is that yeast topo II acts on the unbound region only, which means that there is simply much less DNA available to it in the “chromatinized” experiments, but this fact is not mentioned.

This point brought out by the Referee is also consistent with what we envision might occur even in our assays – topo binds to naked DNA between nucleosomes. We have revised the sentence on page 14 to convey this idea better.

15. In the text, “we again used the repeated winding experiment (Fig. 1b) but with chromatin fibers containing ~50 nucleosomes instead of naked DNA (Fig. 3b)” It is not immediately obvious from this statement or from the Methods which DNA substrate was used. Presumably the 64mer substrate

containing tandem repeats of a nuclear positioning sequence was used. Firstly, the authors should explicitly mention in the main body of the text exactly which substrates were used for which experiments. Secondly, it appears that two very different DNA substrates were used for the naked DNA experiments and the chromatin experiments. The nuclear positioning sequence that is repeated in the 64mer DNA used for the chromatin experiments will affect the properties and behavior of DNA. The naked DNA experiments should be repeated using the 64mer substrate otherwise the comparison is not valid. Sequence and length effects are unlikely to explain all the differences but cannot be ruled out. The lengths of the DNA being used is not given in the text but should be. Later “~50 nucleosomes” are added to this DNA but still the length of DNA is not given in the text. One must dig through the Methods section and then the figure legends to try to figure out what was compared to what. It seems that the chromatinized DNA used was never tested “naked.” But that should be the only DNA compared. Both length and sequence are changing in the comparison, yet this is not acknowledged. For the experiments with “chromatinized DNA”, the comparator (non-chomatinized DNA) must be treated the same way (same DNA sequence and length and torsion put in both positive and negative directions).

The Referee is correct that we used a DNA template containing a non-repeating DNA sequence for the naked DNA experiments and the 64mer DNA template containing 64 repeats of nucleosome positioning element 601 for the chromatin experiments. We apologize for the lack of clarity on this. We now explicitly mention the template type and length on the figure captions of the revised manuscript.

Following the suggestions from the Referee, we have repeated the experiments shown using a naked 64mer DNA substrate under both (+) and (-) winding directions. We have now shown these data in Fig. 6 of the revised manuscript to allow a direct comparison with chromatinized DNA. These data show similar topo II activity lifetime and processivity on the naked 64mer DNA template as on the random DNA template (compare Fig. 2c and 2d with Fig. 6b and 6c). Furthermore, we did not detect any obvious chirality dependence. These new data further strengthen our original conclusion that chromatinization enhances topo II activity lifetime/processivity on DNA under low torsional stress.

16. In the Method section “active tethers” are mentioned, but it is not clear how the method that attaches the DNAs to the slides would be “active” or “inactive.” They are tethers. Perhaps “active” is meant to indicate that it is bound to DNA? Or that it is the one being assessed by the researchers at the given time?

We agree that the term “active tethers” is confusing without a clear definition. We refer to a tether being active if its supercoiling is relaxed by topo II. We have now clarified this on page 27 of the revised Methods.

17. The assumption in the manuscript is that addition of nucleosomes increases DNA crossovers. But there are too many other variables changed to make that assumption. How do the authors rule out the possibility that nucleosomes interact with the yeast topo II (what biochemists refer to as “the BSA effect” because addition of BSA seems to enable/enhance every biochemical reaction)? Already the DNA length and sequence used for the “naked DNA” results used to generate the other rates to which the chromatinized DNA is compared to differ, but this length additionally changes when bound to ~50 nucleosomes but possible ramifications of this major change in the experimental set up is not mentioned.

We understand this comment from this Referee. The statement that nucleosomes increase DNA

crossovers is supported by a large body of literature, such as the work of Ariel Prunell (De Lucia et al., JMB, 1999; Prunell and Sivolob, New Compr. Biochem., 2004; Sivolob and Prunell, Philosophical Transactions of the Royal Society of London Series A: Mathematical, Physical and Engineering Sciences, 2004), but also from many others. Thus, this statement is not an assumption. Although we cannot exclude other possibilities, an increase in DNA crossovers by nucleosomes provides the simplest explanation of our data. In addition, there is limited evidence for direct interaction of yeast topo II with nucleosomes. The DNA sequence also does not play an important role as evidenced by our new data using the 64mer template, now shown in Fig. 6 of the revised manuscript.

18. Referring to the 12,688 bp double-stranded DNA (or later the 64 repeats of 601 NPE and 500 bp adaptors) as a “trunk” is unnecessary and oddly confusing.

We agree with the Referee. In the revised manuscript, we have replaced “trunk” with “center segment”.

19. How do the researchers know and how did they assess that two “flushes” of “5x chamber volume of buffer” is enough to wash away unbound enzyme? Is the efficiency of this wash assessed? If not, then the sentence “These two washes minimize further topo II binding events during measurements” should be changed to “might/are hoped to minimize further topo II binding events.”

We thank the Referee for this suggestion. We have also changed the word “minimize” to “reduce” to make the statement more accurate.

20. From the sentence, “If the number of topo II bound to a tether follows a Poisson distribution, this would correspond to ~90-95% of the active fraction of tethers being bound to a single topo II,” is there any published evidence that yeast topo II binds DNA with a Poisson distribution? If so, a reference(s) should be added. If not, is there any evidence of the opposite for the yeast or any topo II enzyme? Is there evidence that it aggregates or cooperatively binds DNA? If so, then a Poisson distribution is a bad assumption. That it is a bad assumption might be supported by the observation that 27% of the ~10–15% DNAs (not tethers as stated; yeast topo II is surely not interacting or binding to the tethers but to the DNA tethered to the slide by the tethers) is being relaxed 2–3 time faster. But that is assuming that each enzyme binding would be additive in activity. What if the 2–3 fold faster rates seen in nearly 1/3 of the experiments is the true rate and there are two states of topo II—a normal active state and a not as active state and only the not as active state was measured? I struggle to understand, given the experimental set-up, how two or three enzymes acting would yield a 2–3 fold faster rate. It would seem to me that a rate-limiting single enzyme activity would be the read out of the experiment. This section indicated a lack of critical evaluation by the researchers and, perhaps, a too rapid dismissal of a significant portion of the data.

We thank the Referee for giving our interpretation a critical review. As the Referee noted, the Poisson distribution may be a bad assumption, as we are unaware of any published evidence that yeast topo II binds with a Poisson distribution. Therefore, we have worked on a different method to characterize the probability of another topo II molecule relaxing DNA during the wait time of the experiments, which we summarize below:

- 1) First, we used a very low concentration of topo II (0.5 pM). Note that previous, accepted single-molecule studies typically worked with much higher topo II concentrations (in the nM to μ M range). Moreover, although it is a standard practice in the single-molecule community to use the dilution method to achieve a single-molecule condition, we did not find this practice satisfying. In addition to

dilution, we have carried out the following characterizations to quantitatively understand the validity and limitations of this approach.

- 2) Second, we flushed the sample chamber to remove free topo II in solution in the following way: After 2 min of incubation of topo II, we flushed the sample chamber using 5x chamber volume of the topo reaction buffer without topo II, waited for 5 min, and then repeated the flushing. These two washes reduce further topo II binding events during measurements. After this process, typically ~10-15% of our tethers showed topo II relaxation activity when wound down, which we defined as "active." Because of this low probability of a tether being acted upon, we could assume that the activity we did see was most likely from a single topo II molecule. Furthermore, we found that active tether fraction decreased from 70% to 10-15% with the two additional washing steps, indicating that the washes are effective at removing free topo II.
- 3) However, during the course of the subsequent measurement, an additional topo II molecule, which could be a consequence of any incomplete flushing, could become bound to these tethers and relax them. We initially estimated this occurrence assuming topo II binding followed a Poisson distribution. Given the concerns raised by the Referee, which we also shared, we worked on finding an alternative approach to make this estimate using the following method during the process of manuscript revision. We performed the conventional "wind down and wait" control experiment and monitored the tethers that were initially inactive but became active later in the measurements. We found this probability to be $\sim 17\pm 5\%$ over 30 minutes. We think this is a more accurate method of estimation. These results show that our measurements should be predominately from a single topo II.
- 4) For experiments using the constant-extension method, we were able to impose more stringent selection criteria to ensure a single topo enzyme condition. In this method, the activity of an additional topo II molecule caused an abrupt doubling in the relaxation rate. Thus, we could readily identify and exclude those traces from further analysis.

In the revised manuscript, we have expanded the discussion to include these points under Methods on pages 27-28 and provided a brief discussion in the main text on page 5.

The Referee proposed an intriguing possibility that there could be different states of topo II (a normal active state and a not-so-active state). Although we cannot completely exclude this possibility, our data may be explained by considering only a single active state. We observed a predominately unimodal rate distribution under low topo concentrations when only a small fraction of the DNA tethers showed topo relaxation activities (Supplementary Figs. 1e and 3b; Fig. 7). When we increased the topo concentrations, we found that higher rates appeared at an integer multiple of this value with a concurrent increase in the fraction of the DNA tethers showing topo activity. These observations are consistent with our observed rate being the normal rate of a single topo molecule.

21. Given that the length of DNA is known (although not clear to the readers in the manuscript) and the number of turns is known, why isn't there an attempt to quantify degree of supercoiling, or at least relative supercoiling (normalized for length)? It might not be absolute, but it would be highly quantitative relatively. Being more quantitative about degree of supercoiling, at least for the "naked" DNA experiments would be useful and allow quantitative comparisons to be made. If this cannot be done, it must be acknowledged. With the histone bound DNAs, quantification is more challenging, especially because it appears that histones allow "absorption of + supercoiling." In the absence of

such quantitation, however, the authors cannot compare “chromatinized” to “non-chromatinized” DNA. There are too many other variables (different “trunks” used, different DNA sequence, different DNA lengths, shorter effective DNA length when chromatinized and thus less DNA available to enzyme, possible protein-topo II interactions, and more) for this comparison to be made. That all the experimental variables affecting the experiment were not acknowledged by the researchers is bothersome, again indicating a lack of critical thought.

We very much appreciate this insightful comment from the Referee. Since the focus of this manuscript is on how topo II activity responds to torsional stress, we must define torsional stress. Torsional stress is characterized by torque, which measures the resistance to twisting. Torque is a more appropriate parameter than the degree of supercoiling for this work since the degree of supercoiling only indicates the extra turns introduced and does not provide a direct measure of the difficulty of twisting DNA. It is worth mentioning that our lab has pioneered many unique approaches to making torque measurements. Recently, we measured the torque required to twist naked DNA and to twist a plectoneme under different forces (Gao et al., PRL, 2021) and the torque needed to twist a chromatin fiber (Le et al., Cell, 2019), all in the same buffer used in this manuscript (the topo reaction buffer).

Therefore, we have added a new Fig. 1 with data taken from our angular optical trap (AOT) to show direct torsional response of DNA to supercoiling. We have also added the relevant torque plots in the revised Fig. 4b (torque on pre-buckled DNA under different forces) and Fig. 5a (torque on a chromatin fiber). These additions quantitatively define the torsional state of DNA and allow identification of torsion for each experimental condition.

22. The text states, “Taken together, our results suggest topo II dynamically senses torsional stress in DNA and adjusts its activity accordingly (Fig. 5)”. But figure 5 does not indicate how topo may sense torsional stress. It shows how topoisomerase may recognize plectonemes. Plectoneme formation reduces torsional stress. It is the plectonemes that are being recognized (the consequence of torsional strain), not torsional stress itself.

This statement has been removed in the revised manuscript.

23. There were a few sentences that contained editing/splicing errors problematic enough to make the meaning obscured. This is one example: “This released the “inner-turn” DNA (interacting primarily with the H3/H4 tetramer) was released from the histone core octamers, which is ~75 bp in length for each nucleosome.”

We thank the Referee for spotting the typo, which we have corrected. We have also gone through an additional round of editing to fix other errors of this kind.

24. In the text “based on source code available on Omar Saleh’s website”, who is Omar Saleh? The URL should be provided.

We have revised this sentence to provide the citation to the source code instead.

25. Equally odd were two comments in the editorial policy checklist: “We will include individual data points upon revision of the manuscript. Our current plots show the mean with an n in the range of 9 to 16.” And “We will update our plots to show the data distribution upon revision. Our current plots show the mean with a clearly defined error bar.” Why not just do it right in the first submission?

We have revised the manuscript to follow the checklist.

Overall, the study potentially provides some interesting observations. But in its current form it is let down by the issues described above.

We greatly appreciate the detailed and constructive comments from this Referee. We hope our revised manuscript has fully addressed the concerns from this Referee.

REVIEWERS' COMMENTS

Reviewer #1 (Remarks to the Author):

I remain positive about this manuscript (as stated in the first round). The authors have carefully addressed the points that were raised by me (and as far as I can tell also the other reviewers). This has really helped clarity and quality. I fully support publication of the manuscript in Nature Comm, in its current form,

Reviewer #2 (Remarks to the Author):

The authors have largely addressed my initial concerns with the manuscript. I appreciate the extensive efforts they put in to include additional data, simulations, and provide a more coherent explanation and rationale for their claims of measuring the activity of topoisomerase II as a function of torsion in the manuscript. The revised Figure 1 is particularly illuminating and informative. Despite my overall enthusiasm for the greatly improved revised manuscript, there remain a few points that the authors should address prior to acceptance or publication.

1. Although I appreciate the point that the authors make in the rebuttal concerning their definition of torsion and torsion-dependent changes in type II topoisomerase activity, I think that the manuscript remains unclear and confusing due to the understanding of torsion in the field. At a constant force, the authors are correct that the torsion will increase before the DNA buckles to form a plectoneme. However, as they correctly state in the manuscript, the constant torsion in the plectoneme depends on the applied force. For this reason references to changes in activity and processivity with torsion in the manuscript are confusing since the changes are associated more accurately with changes in the state of the DNA, which depend on the torsion but also on the applied force. The authors make the point clearly in the rebuttal letter, but throughout the manuscript there are vestiges of the highly confusing statement that the activity changes as a function of torsion, which is true at the given force they are working under, but not generally true. Perhaps if they defined this or stipulated the torsion at a given force then it would make their statements more clear. Alternatively, they could refer to the phase of the DNA, which is the largest factor in altering the activity and processivity of the type II topoisomerases.

2. P7. In the revised manuscript, the authors state that turns were rapidly added at 40 turns/s. This is a quite fast rotation rate for magnetic tweezers studies and, depending on the details of the magnetic tweezers system and the drag on the bead, could result in "slipping" of the bead, which would result in fewer turns being applied to the bead than expected. I expect that the authors have established that the bead rotation remains locked to the magnet rotation in a 1-to-1 manner at this high rotation rate, but it would be helpful to demonstrate this in a supplemental figure to allay this concern.

3. P9. The authors make the claim that the increased processivity of the type II topoisomerases that they observe is due to the technique employed in this work in which the magnetic bead is constantly rotated to maintain the length of the plectoneme on which the enzyme is working. The claim that previous methods observed lower processivity due to technical or inherent methodological flaws, is problematic for two reasons. It seems that previous measurements of processivity also maintained the DNA in a plectonemic state, but compensated for the relaxation activity of the topoisomerase in a discontinuous manner rather than immediately compensating for the relaxation of the plectoneme by the topoisomerase as was done in this work. It would seem from the conclusions of the current work that one would expect similar activity and processivity on a plectoneme of any length, and that the discontinuous rewinding should not result in lower processivity measurements under this reasonable assumption. If the authors feel strongly that the continuous feedback method produces different results than the discontinuous feedback method, they could easily demonstrate this effect by performing both measurements. However, this possibility raises an important consideration about the

measurements. If the continuous feedback method indeed results in significantly higher processivity measurements than the discontinuous feedback method to maintain the DNA in a plectonemic state, then it would suggest that something other than the plectonemic state of the DNA impacts the processivity. This would further raise the concern that the instantaneous rotation of the magnetic bead and reforming of the plectoneme may artificially increase the processivity of the topoisomerase. The underlying assumption in the measurements of processivity is that higher is necessarily better and closer to physiological, but this really should be justified, by for example slowing down the feedback to maintain the plectoneme in a discontinuous manner as done previously. If the results are dramatically different when measured with the same enzymes, under identical conditions, then the authors can justifiably claim that there is a difference in the processivity measured with the two similar but subtly different techniques. Such a finding would require a more detailed analysis of the two methods and could reveal interesting dynamic effects on the processivity of the type II topoisomerases. It is also possible and more likely that the differences in processivity arise from different enzyme preparations and buffer conditions, which are known to dramatically affect processivity. Since the authors are insistent that the instantaneous assay is technically superior and provides a more accurate and faithful measure of the processivity, they should back up this claim with a direct head-to-head comparison that they are uniquely positioned to perform. Unfortunately, until this apparent discrepancy is resolved, it is unclear which approach provides a better measure of the intrinsic processivity of the type II topoisomerases.

4. P13. The sentence "Interestingly, at a given supercoiling state, topo II relaxation rate slightly decreased when DNA torsion (torque) increased with an increase in force from 0.5 pN to 1.6 pN, suggesting that changes in torsion alone are not sufficient to explain the variation in topo II relaxation rate." is confusing and seemingly contradictory. It seems that the results indicate that torsion does influence the relaxation rate – that is precisely what the data is indicating. I think that the confusion relates to the continued issue with the conflation of torsion and buckling state. In the pre-buckled state the increased force will result in increased torsion and as the authors observe, a subtle change in activity, which is distinct from the much larger change in activity associated with entering the buckling regime at a slightly higher torsion, dependent on the applied force.

5. P14. Can the authors distinguish between the imposed bend and native bending models? The fact that both models fit the data apparently equally as well seems surprising. Is there any additional data from these measurements or the literature that could distinguish these two models?

6. P15. "high torsional stress" the meaning of this is ambiguous in this context. I think that the authors mean "buckled" but this could mean unbuckled under high applied force. This is another example of the ambiguity in the term "torsion" to refer to different phases of DNA.

7. P19. " High torsional stress" once again this is ambiguous as stated. The authors are in fact referring to buckled plectonemic DNA, which occurs at higher torsion than unbuckled DNA, but the same level of torsional stress could be achieved in unbuckled DNA at a higher force.

Once the authors address these few remaining concerns then the manuscript would be suitable for publication.

Reviewer #3 (Remarks to the Author):

This is a nice study. The new data added strengthen the conclusions and are satisfactorily robust. I'm glad the authors found our comments useful.

Response to Reviewers

Re: NCOMMS-23-05291
"Chromatinization Modulates Topoisomerase II Processivity" by Lee et al.

We greatly appreciate the helpful comments and suggestions from the three reviewers and the Editor. Reviewer #1 and Reviewer #3 are satisfied with our responses. Therefore, we will focus on comments from Reviewer #2. Below, we provide a detailed, point-by-point response to each comment from the three referees, with each comment (bold) followed by our response (not bold). Note that in the revised manuscript, we have marked the suggested changes in red.

Reviewer #1 (Remarks to the Author):

I remain positive about this manuscript (as stated in the first round). The authors have carefully addressed the points that were raised by me (and as far as I can tell also the other reviewers). This has really helped clarity and quality. I fully support publication of the manuscript in Nature Comm, in its current form.

We very much appreciate all of the helpful comments from this Reviewer throughout the review process, and we thank the Reviewer for supporting the publication of the manuscript in *Nature Communications*.

Reviewer #2 (Remarks to the Author):

The authors have largely addressed my initial concerns with the manuscript. I appreciate the extensive efforts they put in to include additional data, simulations, and provide a more coherent explanation and rational for their claims of measuring the activity of topo II as a function of torsion in the manuscript. The revised Figure 1 is particularly illuminating and informative. Despite my overall enthusiasm for the greatly improved revised manuscript, there remain a few points that the authors should address prior to acceptance or publication.

We are glad that the Reviewer finds Figure 1 informative. We anticipate other readers may also find it helpful.

1. Although I appreciate the point that the authors make in the rebuttal concerning their definition of torsion and torsion-dependent changes in type II topoisomerase activity, I think that the manuscript remains unclear and confusing due to the understanding of torsion in the field. At a constant force, the authors are correct that the torsion will increase before the DNA buckles to form a plectoneme. However, as they correctly state in the manuscript, the constant torsion in the plectoneme depends on the applied force. For this reason references to changes in activity and processivity with torsion in the manuscript are confusing since the changes are associated more accurately with changes in the state of the DNA, which depend on the torsion but also on the applied force. The authors make the point clearly in the rebuttal letter, but throughout the manuscript there are vestiges of the highly confusing statement that the activity changes as a function of torsion, which is true at the given force they are working under, but not generally true. Perhaps if they defined this or stipulated the torsion at a given force then it would make their statements more clear. Alternatively, they could refer to the phase of the DNA, which is the largest factor in altering the activity and processivity of the type II

topoisomerases.

As the Reviewer noted, torsion in DNA is indeed a complex topic as it depends on the superhelical density of DNA and force. Indeed, our lab has spent a great effort to characterize the force-torque phase diagram of DNA (Duefel et al., Nat. Methods, 2007; Forth et al., PRL, 2008; Sheinin et al., PRL, 2011; Gao et al., PRL, 2021).

Upon reading the comments from this Reviewer, we have gone through the manuscript and made the following changes. (1) We have now explicitly indicated the force whenever relevant, as suggested by the Reviewer. (2) We have changed “high torsional stress” to “buckled” and “low torsional stress” to “pre-buckled,” as suggested by the Reviewer.

2. P7. In the revised manuscript, the authors state that turns were rapidly added at 40 turns/s. This is a quite fast rotation rate for magnetic tweezers studies and, depending on the details of the magnetic tweezers system and the drag on the bead, could result in “slipping” of the bead, which would result in fewer turns being applied to the bead than expected. I expect that the authors have established that the bead rotation remains locked to the magnet rotation in a 1-to-1 manner at this high rotation rate, but it would be helpful to demonstrate this in a supplemental figure to allay this concern.

We have routinely verified that the magnetic beads do not slip when rotated at 40 turns/s. A relevant control experiment is now shown as the new Supplementary Fig. 2b, where we rotated the magnets to add +30 turns at 40 turns/s and verified that +30 turns were added to the DNA, demonstrating that the bead rotation indeed remains locked to the magnet rotation even at this high rotation rate.

3. P9. The authors make the claim that the increased processivity of the type II topoisomerases that they observe is due to the technique employed in this work in which the magnetic bead is constantly rotated to maintain the length of the plectoneme on which the enzyme is working. The claim that previous methods observed lower processivity due to technical or inherent methodological flaws, is problematic for two reasons. It seems that previous measurements of processivity also maintained the DNA in a plectonemic state, but compensated for the relaxation activity of the topoisomerase in a discontinuous manner rather than immediately compensating for the relaxation of the plectoneme by the topoisomerase as was done in this work. It would seem from the conclusions of the current work that one would expect similar activity and processivity on a plectoneme of any length, and that the discontinuous rewinding should not result in lower processivity measurements under this reasonable assumption. If the authors feel strongly that the continuous feedback method produces different results than the discontinuous feedback method, they could easily demonstrate this effect by performing both measurements. However, this possibility raises an important consideration about the measurements. If the continuous feedback method indeed results in significantly higher processivity measurements than the discontinuous feedback method to maintain the DNA in a plectonemic state, then it would suggest that something other than the plectonemic state of the DNA impacts the processivity. This would further raise the concern that the instantaneous rotation of the magnetic bead and reforming of the plectoneme may artificially increase the processivity of the topoisomerase. The underlying assumption in the measurements of processivity is that higher is necessarily better and closer to physiological, but this really should be justified, by for example slowing down the feedback to maintain the plectoneme in a discontinuous manner as done previously. If the results are dramatically different when measured with the same enzymes, under identical conditions, then the authors can justifiably claim that there is a difference in the processivity measured with the two similar but subtly different techniques. Such a finding would require a more detailed analysis of the

two methods and could reveal interesting dynamic effects on the processivity of the type II topoisomerases. It is also possible and more likely that the differences in processivity arise from different enzyme preparations and buffer conditions, which are known to dramatically affect processivity. Since the authors are insistent that the instantaneous assay is technically superior and provides a more accurate and faithful measure of the processivity, they should back up this claim with a direct head-to-head comparison that they are uniquely positioned to perform. Unfortunately, until this apparent discrepancy is resolved, it is unclear which approach provides a better measure of the intrinsic processivity of the type II topoisomerases.

Our existing data in Fig. 1 effectively directly compare our constant-extension method and the previous manual reset method employed by Seol et al. (JBC, 2013). Fig. 1a shows our new constant-extension method, whereas Fig. 1b is the repeated winding experiments, which bears resemblance to that method used by Seol et al. In the repeated winding experiments, we started the measurement on buckled DNA, allowed topo II to relax into the pre-buckled region, and subsequently brought the DNA back to the buckled region by mechanically adding additional turns.

Figs. 1c and 1d demonstrate that once DNA became pre-buckled, topo II processivity was reduced greatly, removing only about 30 turns of the pre-buckled DNA before ceasing activity. This corresponds to a 200 X reduction in processivity from that of the buckled DNA. If we sum all turns relaxed from when the DNA was buckled to when the DNA was unbuckled in a repeated winding experiment, the total processivity would be 24 turns. This processivity is less than that of Seol et al. and is far below the 6000 turns of processivity on buckled DNA of our work. These results suggest that keeping the DNA in the buckled state at all times is essential to sustain a high topo processivity.

That being said, it is not possible to fully exclude that the differences in processivity could have arisen from different enzyme preparations and buffer conditions. We have modified our discussion to mention these possibilities on page 9 of the revised manuscript.

4. P13. The sentence “Interestingly, at a given supercoiling state, topo II relaxation rate slightly decreased when DNA torsion (torque) increased with an increase in force from 0.5 pN to 1.6 pN, suggesting that changes in torsion alone are not sufficient to explain the variation in topo II relaxation rate.” is confusing and seemingly contradictory. It seems that the results indicate that torsion does influence the relaxation rate – that is precisely what the data is indicating. I think that the confusion relates to the continued issue with the conflation of torsion and buckling state. In the pre-buckled state the increased force will result in increased torsion and as the authors observe, a subtle change in activity, which is distinct from the much larger change in activity associated with entering the buckling regime at a slightly higher torsion, dependent on the applied force.

We agree that this sentence can be confusing. We have now reworded this sentence to state that topo II relaxation rate is more sensitive to DNA supercoiling state than torsion alone on page 12 of the revised manuscript.

5. P14. Can the authors distinguish between the imposed bend and native bending models? The fact that both models fit the data apparently equally as well seems surprising. Is there any additional data from these measurements or the literature that could distinguish these two models?

Our DNA looping simulation of DNA with an intrinsic bend shows a somewhat increased rate of DNA loop formation. This effect was reflected as a reduced value of $k_{1/2}$ in our model, as we previously

described in the Methods section and is now also discussed in the main text on page 13 of the revised manuscript.

Differentiating the contributions between an intrinsic bend and a native bend in DNA is beyond the scope of the manuscript.

6. P15. “high torsional stress” the meaning of this is ambiguous in this context. I think that the authors mean “buckled” but this could mean unbuckled under high applied force. This is another example of the ambiguity in the term “torsion” to refer to different phases of DNA.

We have changed “high torsional stress” to “buckled” as suggested by the Reviewer.

7. P19. ” High torsional stress” once again this is ambiguous as stated. The authors are in fact referring to buckled plectonemic DNA, which occurs at higher torsion than unbuckled DNA, but the same level of torsional stress could be achieved in unbuckled DNA at a higher force.

We have also changed “high torsional stress” to “buckled” as suggested by the Reviewer.

Once the authors address these few remaining concerns then the manuscript would be suitable for publication.

Reviewer #3 (Remarks to the Author):

This is a nice study. The new data added strengthen the conclusions and are satisfactorily robust. I'm glad the authors found our comments useful.

We are delighted that we have addressed all the concerns of this Reviewer. We want to thank this Reviewer again for their helpful comments once again.